# An Improved Analysis of Training Over-parameterized Deep Neural Networks

**Difan Zou**
Department of Computer Science
University of California, Los Angeles
Los Angeles, CA 90095
knowzou@cs.ucla.edu

**Quanquan Gu**
Department of Computer Science
University of California, Los Angeles
Los Angeles, CA 90095
qgu@cs.ucla.edu

## Abstract

A recent line of research has shown that gradient-based algorithms with random initialization can converge to the global minima of the training loss for over-parameterized (i.e., sufficiently wide) deep neural networks. However, the condition on the width of the neural network to ensure the global convergence is very stringent, which is often a high-degree polynomial in the training sample size $n$ (e.g., $O(n^{24})$). In this paper, we provide an improved analysis of the global convergence of (stochastic) gradient descent for training deep neural networks, which only requires a milder over-parameterization condition than previous work in terms of the training sample size and other problem-dependent parameters. The main technical contributions of our analysis include (a) a tighter gradient lower bound that leads to a faster convergence of the algorithm, and (b) a sharper characterization of the trajectory length of the algorithm. By specializing our result to two-layer (i.e., one-hidden-layer) neural networks, it also provides a milder over-parameterization condition than the best-known result in prior work.

## 1 Introduction

Recent study [20] has revealed that deep neural networks trained by gradient-based algorithms can fit training data with random labels and achieve zero training error. Since the loss landscape of training deep neural network is highly nonconvex or even nonsmooth, conventional optimization theory cannot explain why gradient descent (GD) and stochastic gradient descent (SGD) can find the global minimum of the loss function (i.e., achieving zero training error). To better understand the training of neural networks, there is a line of research [18, 5, 10, 16, 23, 8, 22, 12] studying two-layer (i.e., one-hidden-layer) neural networks, where it assumes there exists a teacher network (i.e., an underlying ground-truth network) generating the output given the input, and casts neural network learning as weight matrix recovery for the teacher network. However, these studies not only make strong assumptions on the training data (existence of ground-truth network with the same architecture as the learned network), but also need special initialization methods that are very different from the commonly used initialization method [13] in practice. Li and Liang [15], Du et al. [11] advanced this line of research by proving that under much milder assumptions on the training data, (stochastic) gradient descent can attain a global convergence for training over-parameterized (i.e.,sufficiently wide) two-layer ReLU network with widely used random initialization method [13]. More recently, Allen-Zhu et al. [2], Du et al. [9], Zou et al. [24] generalized the global convergence results from two-layer networks to deep neural networks. However, there is a huge gap between the theory and practice since all these work Li and Liang [15], Du et al. [11], Allen-Zhu et al. [2], Du et al. [9], Zou et al. [24] require unrealistic over-parameterization conditions on the width of neural networks, especially for deep networks. In specific, in order to establish the global convergence for training two-layer ReLU networks, Du et al. [11] requires the network width, i.e., number of hidden

nodes, to be at least $\Omega(n^6/\lambda_0^4)$, where $n$ is the training sample size and $\lambda_0$ is the smallest eigenvalue of the so-called Gram matrix defined in Du et al. [11], which is essentially the neural tangent kernel [14, 7] on the training data. Under the same assumption on the training data, Wu et al. [19] improved the iteration complexity of GD in Du et al. [11] from $O\big(n^2\log(1/\epsilon)/\lambda_0^2\big)$ to $O\big(n\log(1/\epsilon)/\lambda_0\big)$ and Oymak and Soltanolkotabi [17] improved the over-parameterization condition to $\Omega(n\|\mathbf{X}\|_2^6/\lambda_0^4)$, where $\epsilon$ is the target error and $\mathbf{X} \in \mathbb{R}^{n\times d}$ is the input data matrix. For deep ReLU networks, the best known result was established in Allen-Zhu et al. [2], which requires the network width to be at least $\widetilde{\Omega}(kn^{24}L^{12}\phi^{-8})$[1] to ensure the global convergence of GD and SGD, where $L$ is the number of hidden layers, $\phi$ is the minimum data separation distance and $k$ is the output dimension.

This paper continues the line of research, and improves the over-parameterization condition and the global convergence rate of (stochastic) gradient descent for training deep neural networks. In specific, under the same setting as in Allen-Zhu et al. [2], we prove faster global convergence rates for both GD and SGD under a significantly milder condition on the neural network width. Furthermore, when specializing our result to two-layer ReLU networks, it also outperforms the best-known result proved in Oymak and Soltanolkotabi [17]. The improvement in our result is due to the following two innovative proof techniques: (a) a tighter gradient lower bound, which leads to a faster rate of convergence for GD/SGD; and (b) a sharper characterization of the trajectory length for GD/SGD until convergence.

We highlight our main contributions as follows:

- We show that, with Gaussian random initialization [13] on each layer, when the number of hidden nodes per layer is $\widetilde{\Omega}\big(kn^8L^{12}\phi^{-4}\big)$, GD can achieve $\epsilon$ training loss within $\widetilde{O}\big(n^2L^2\log(1/\epsilon)\phi^{-1}\big)$ iterations, where $L$ is the number of hidden layers, $\phi$ is the minimum data separation distance, $n$ is the number of training examples, and $k$ is the output dimension. Compared with the state-of-the-art result [2], our over-parameterization condition is milder by a factor of $\widetilde{\Omega}(n^{16}\phi^{-4})$, and our iteration complexity is better by a factor of $\widetilde{O}(n^4\phi^{-1})$.

- We also prove a similar convergence result for SGD. We show that with Gaussian random initialization [13] on each layer, when the number of hidden nodes per layer is $\widetilde{\Omega}\big(kn^{17}L^{12}B^{-4}\phi^{-8}\big)$, SGD can achieve $\epsilon$ expected training loss within $\widetilde{O}\big(n^5\log(1/\epsilon)B^{-1}\phi^{-2}\big)$ iterations, where $B$ is the minibatch size of SGD. Compared with the corresponding results in Allen-Zhu et al. [2], our results are strictly better by a factor of $\widetilde{\Omega}(n^7B^5)$ and $\widetilde{O}(n^2)$ respectively regarding over-parameterization condition and iteration complexity.

- When specializing our results of training deep ReLU networks with GD to two-layer ReLU networks, it also outperforms the corresponding results [11, 19, 17]. In addition, for training two-layer ReLU networks with SGD, we are able to show much better result than training deep ReLU networks with SGD.

For the ease of comparison, we summarize the best-known results [11, 2, 9, 19, 17] of training overparameterized neural networks with GD and compare with them in terms of over-parameterization condition and iteration complexity in Table 1. We will show in Section 3 that, under the assumption that all training data points have unit $\ell_2$ norm, which is the common assumption made in all these work [11, 2, 9, 19, 17], $\lambda_0 > 0$ is equivalent to the fact that all training data are separated by some distance $\phi$, and we have $\lambda_0 = O(n^{-2}\phi)$ [17]. Substituting $\lambda_0 = \Omega(n^{-2}\phi)$ into Table 1, it is evident that our result outperforms all the other results under the same assumptions.

**Notation** For scalars, vectors and matrices, we use lower case, lower case bold face, and upper case bold face letters to denote them respectively. For a positive integer, we denote by $[k]$ the set $\{1,\ldots,k\}$. For a vector $\mathbf{x} = (x_1,\ldots,x_d)^\top$ and a positive integer $p$, we denote by $\|\mathbf{x}\|_p = \big(\sum_{i=1}^d |x_i|^p\big)^{1/p}$ the $\ell_p$ norm of $\mathbf{x}$. In addition, we denote by $\|\mathbf{x}\|_\infty = \max_{i=1,\ldots,d}|x_i|$ the $\ell_\infty$ norm of $\mathbf{x}$, and $\|\mathbf{x}\|_0 = |\{x_i : x_i \neq 0, i = 1,\ldots,d\}|$ the $\ell_0$ norm of $\mathbf{x}$. For a matrix $\mathbf{A} \in \mathbb{R}^{m\times n}$, we denote by $\|\mathbf{A}\|_F$ the Frobenius norm of $\mathbf{A}$, $\|\mathbf{A}\|_2$ the spectral norm (maximum singular value), $\lambda_{\min}(\mathbf{A})$ the smallest singular value, $\|\mathbf{A}\|_0$ the number of nonzero entries, and $\|\mathbf{A}\|_{2,\infty}$ the maximum $\ell_2$ norm over all row vectors, i.e., $\|\mathbf{A}\|_{2,\infty} = \max_{i=1,\ldots,m}\|\mathbf{A}_{i*}\|_2$. For a collection of matrices $\mathbf{W} = \{\mathbf{W}_1,\ldots,\mathbf{W}_L\}$, we denote $\|\mathbf{W}\|_F = \sqrt{\sum_{l=1}^L \|\mathbf{W}_l\|_F^2}$, $\|\mathbf{W}\|_2 = \max_{l\in[L]}\|\mathbf{W}_l\|_2$ and

Table 1: Over-parameterization conditions and iteration complexities of GD for training overparameterized neural networks. $\mathbf{K}^{(L)}$ is the Gram matrix for $L$-hidden-layer neural network [9]. Note that the dimension of the output is $k = 1$ in Du et al. [11, 9], Wu et al. [19], Oymak and Soltanolkotabi [17].

| | Over-para. condition | Iteration complexity | Deep? | ReLU? |
|---|---|---|---|---|
| Du et al. [11] | $\Omega\left(\frac{n^6}{\lambda_0^4}\right)$ | $O\left(\frac{n^2\log(1/\epsilon)}{\lambda_0^2}\right)$ | no | yes |
| Wu et al. [19] | $\Omega\left(\frac{n^6}{\lambda_0^4}\right)$ | $O\left(\frac{n\log(1/\epsilon)}{\lambda_0}\right)$ | no | yes |
| Oymak and Soltanolkotabi [17] | $\Omega\left(\frac{n\|\mathbf{X}\|_2^6}{\lambda_0^4}\right)$ | $O\left(\frac{\|\mathbf{X}\|_2^2\log(1/\epsilon)}{\lambda_0}\right)$ | no | yes |
| Du et al. [9] | $\Omega\left(\frac{2^{O(L)}\cdot n^4}{\lambda_{\min}^4(\mathbf{K}^{(L)})}\right)$ | $O\left(\frac{2^{O(L)}\cdot n^2\log(1/\epsilon)}{\lambda_{\min}^2(\mathbf{K}^{(L)})}\right)$ | yes | no |
| Allen-Zhu et al. [2] | $\widetilde{\Omega}\left(\frac{kn^{24}L^{12}}{\phi^8}\right)$ | $O\left(\frac{n^6L^2\log(1/\epsilon)}{\phi^2}\right)$ | yes | yes |
| **This paper** | $\widetilde{\Omega}\left(\frac{kn^8L^{12}}{\phi^4}\right)$ | $O\left(\frac{n^2L^2\log(1/\epsilon)}{\phi}\right)$ | yes | yes |

$\|\mathbf{W}\|_{2,\infty} = \max_{l\in[L]}\|\mathbf{W}_l\|_{2,\infty}$. Given two collections of matrices $\widetilde{\mathbf{W}} = \{\widetilde{\mathbf{W}}_1,\ldots,\widetilde{\mathbf{W}}_L\}$ and $\widehat{\mathbf{W}} = \{\widehat{\mathbf{W}}_1,\ldots,\widehat{\mathbf{W}}_L\}$, we define their inner product as $\langle\widetilde{\mathbf{W}},\widehat{\mathbf{W}}\rangle = \sum_{l=1}^L\langle\widetilde{\mathbf{W}}_l,\widehat{\mathbf{W}}_l\rangle$. For two sequences $\{a_n\}$ and $\{b_n\}$, we use $a_n = O(b_n)$ to denote that $a_n \le C_1 b_n$ for some absolute constant $C_1 > 0$, and use $a_n = \Omega(b_n)$ to denote that $a_n \ge C_2 b_n$ for some absolute constant $C_2 > 0$. In addition, we use $\widetilde{O}(\cdot)$ and $\widetilde{\Omega}(\cdot)$ to hide logarithmic factors.

## 2 Problem setup and algorithms

In this section, we introduce the problem setup and the training algorithms.

Following Allen-Zhu et al. [2], we consider the training of an $L$-hidden layer fully connected neural network, which takes $\mathbf{x} \in \mathbb{R}^d$ as input, and outputs $\mathbf{y} \in \mathbb{R}^k$. In specific, the neural network is a vector-valued function $\mathbf{f_W} : \mathbb{R}^d \to \mathbb{R}^k$, which is defined as

$$\mathbf{f_W}(\mathbf{x}) = \mathbf{V}\sigma(\mathbf{W}_L\sigma(\mathbf{W}_{L-1}\cdots\sigma(\mathbf{W}_1\mathbf{x})\cdots)),$$

where $\mathbf{W}_1 \in \mathbb{R}^{m\times d}$, $\mathbf{W}_2,\ldots,\mathbf{W}_L \in \mathbb{R}^{m\times m}$ denote the weight matrices for the hidden layers, and $\mathbf{V} \in \mathbb{R}^{k\times m}$ denotes the weight matrix in the output layer, $\sigma(x) = \max\{0,x\}$ is the entry-wise ReLU activation function. In addition, we denote by $\sigma'(x) = \mathbb{1}(x)$ the derivative of ReLU activation function and $\mathbf{w}_{l,j}$ the weight vector of the $j$-th node in the $l$-th layer.

Given a training set $\{(\mathbf{x}_i,\mathbf{y}_i)\}_{i=1,\ldots,n}$ where $\mathbf{x}_i \in \mathbb{R}^d$ and $\mathbf{y}_i \in \mathbb{R}^k$, the empirical loss function for training the neural network is defined as

$$L(\mathbf{W}) := \frac{1}{n}\sum_{i=1}^n \ell(\widehat{\mathbf{y}}_i,\mathbf{y}_i), \tag{2.1}$$

where $\ell(\cdot,\cdot)$ is the loss function, and $\widehat{\mathbf{y}}_i = \mathbf{f_W}(\mathbf{x}_i)$. In this paper, for the ease of exposition, we follow Allen-Zhu et al. [2], Du et al. [11, 9], Oymak and Soltanolkotabi [17] and consider square loss as follows

$$\ell(\widehat{\mathbf{y}}_i,\mathbf{y}_i) = \frac{1}{2}\|\mathbf{y}_i - \widehat{\mathbf{y}}_i\|_2^2,$$

where $\widehat{\mathbf{y}}_i = \mathbf{f_W}(\mathbf{x}_i) \in \mathbb{R}^k$ denotes the output of the neural network given input $\mathbf{x}_i$. It is worth noting that our result can be easily extended to other loss functions such as cross entropy loss [24] as well.

We will study both gradient descent and stochastic gradient descent as training algorithms, which are displayed in Algorithm 1. For gradient descent, we update the weight matrix $\mathbf{W}_l^{(t)}$ using full partial gradient $\nabla_{\mathbf{W}_l}L(\mathbf{W}^{(t)})$. For stochastic gradient descent, we update the weight matrix $\mathbf{W}_l^{(t)}$ using stochastic partial gradient $1/B\sum_{s\in\mathcal{B}^{(t)}}\nabla_{\mathbf{W}_l}\ell\big(\mathbf{f}_{\mathbf{W}^{(t)}}(\mathbf{x}_s),\mathbf{y}_s\big)$, where $\mathcal{B}^{(t)}$ with $|\mathcal{B}^{(t)}| = B$ denotes the minibatch of training examples at the $t$-th iteration. Both algorithms are initialized in the same

---

**Algorithm 1** (Stochastic) Gradient descent with Gaussian random initialization

---

1: **input:** Training data $\{\mathbf{x}_i, \mathbf{y}_i\}_{i \in [n]}$, step size $\eta$, total number of iterations $T$, minibatch size $B$.

2: **initialization:** For all $l \in [L]$, each row of weight matrix $\mathbf{W}_l^{(0)}$ is independently generated from $N(0, 2/m\mathbf{I})$, each row of $\mathbf{V}$ is independently generated from $N(0, \mathbf{I}/k)$

$\rule{5cm}{0.4pt}$ **Gradient Descent** $\rule{5cm}{0.4pt}$

3: **for** $t = 0, \ldots, T$ **do**
4:     $\mathbf{W}_l^{(t+1)} = \mathbf{W}_l^{(t)} - \eta \nabla_{\mathbf{W}_l} L(\mathbf{W}^{(t)})$ for all $l \in [L]$
5: **end for**
6: **output:** $\{\mathbf{W}_l^{(T)}\}_{l \in [L]}$

$\rule{5cm}{0.4pt}$ **Stochastic Gradient Descent** $\rule{5cm}{0.4pt}$

7: **for** $t = 0, \ldots, T$ **do**
8:     Uniformly sample a minibatch of training data $\mathcal{B}^{(t)} \in [n]$
9:     $\mathbf{W}_l^{(t+1)} = \mathbf{W}_l^{(t)} - \frac{\eta}{B} \sum_{s \in \mathcal{B}^{(t)}} \nabla_{\mathbf{W}_l} \ell\big(\mathbf{f}_{\mathbf{W}^{(t)}}(\mathbf{x}_s), \mathbf{y}_s\big)$ for all $l \in [L]$
10: **end for**
11: **output:** $\{\mathbf{W}_l^{(T)}\}_{l \in [L]}$

---

way as Allen-Zhu et al. [2], which is essentially the initialization method [13] widely used in practice. In the remaining of this paper, we denote by

$$\nabla L(\mathbf{W}^{(t)}) = \{\nabla_{\mathbf{W}_l} L(\mathbf{W}^{(t)})\}_{l \in [L]} \quad \text{and} \quad \nabla \ell\big(\mathbf{f}_{\mathbf{W}^{(t)}}(\mathbf{x}_i), \mathbf{y}_i\big) = \{\nabla_{\mathbf{W}_l} \ell\big(\mathbf{f}_{\mathbf{W}^{(t)}}(\mathbf{x}_i), \mathbf{y}_i\big)\}_{l \in [L]}$$

the collections of all partial gradients of $L(\mathbf{W}^{(t)})$ and $\ell\big(\mathbf{f}_{\mathbf{W}^{(t)}}(\mathbf{x}_i), \mathbf{y}_i\big)$.

## 3 Main theory

In this section, we present our main theoretical results. We make the following assumptions on the training data.

**Assumption 3.1.** For any $\mathbf{x}_i$, it holds that $\|\mathbf{x}_i\|_2 = 1$ and $(\mathbf{x}_i)_d = \mu$, where $\mu$ is an positive constant.

The same assumption has been made in all previous work along this line [9, 2, 24, 17]. Note that requiring the norm of all training examples to be 1 is not essential, and this assumption can be relaxed to be $\|\mathbf{x}_i\|_2$ is lower and upper bounded by some constants.

**Assumption 3.2.** For any two different training data points $\mathbf{x}_i$ and $\mathbf{x}_j$, there exists a positive constant $\phi > 0$ such that $\|\mathbf{x}_i - \mathbf{x}_j\|_2 \geq \phi$.

This assumption has also been made in Allen-Zhu et al. [3, 2], which is essential to guarantee zero training error for deep neural networks. It is a quite mild assumption for the regression problem as studied in this paper. Note that Du et al. [9] made a different assumption on training data, which requires the Gram matrix $\mathbf{K}^{(L)}$ (See their paper for details) defined on the $L$-hidden-layer networks is positive definite. However, their assumption is not easy to verify for neural networks with more than two layers.

Based on Assumptions 3.1 and 3.2, we are able to establish the global convergence rates of GD and SGD for training deep ReLU networks. We start with the result of GD for $L$-hidden-layer networks.

### 3.1 Training $L$-hidden-layer ReLU networks with GD

The global convergence of GD for training deep neural networks is stated in the following theorem.

**Theorem 3.3.** Under Assumptions 3.1 and 3.2, and suppose the number of hidden nodes per layer satisfies

$$m = \Omega\big(kn^8 L^{12} \log^3(m)/\phi^4\big). \tag{3.1}$$

Then if set the step size $\eta = O\big(k/(L^2 m)\big)$, with probability at least $1 - O(n^{-1})$, gradient descent is able to find a point that achieves $\epsilon$ training loss within

$$T = O\big(n^2 L^2 \log(1/\epsilon)/\phi\big)$$

iterations.

**Remark 3.4.** The state-of-the-art results for training deep ReLU network are provided by Allen-Zhu et al. [2], where the authors showed that GD can achieve $\epsilon$-training loss within $O\big(n^6L^2\log(1/\epsilon)/\phi^2\big)$ iterations if the neural network width satisfies $m = \widetilde{\Omega}\big(kn^{24}L^{12}/\phi^8\big)$. As a clear comparison, our result on the iteration complexity is better than theirs by a factor of $O(n^4/\phi)$, and our over-parameterization condition is milder than theirs by a factor of $\widetilde{\Omega}(n^{16}/\phi^4)$. Du et al. [9] also proved the global convergence of GD for training deep neural network with smooth activation functions. As shown in Table 1, the over-parameterization condition and iteration complexity in Du et al. [9] have an exponential dependency on $L$, which is much worse than the polynomial dependency on $L$ as in Allen-Zhu et al. [2] and our result.

We now specialize our results in Theorem 3.3 to two-layer networks by removing the dependency on the number of hidden layers, i.e., $L$. We state this result in the following corollary.

**Corollary 3.5.** Under the same assumptions made in Theorem 3.3. For training two-layer ReLU networks, if set the number of hidden nodes $m = \Omega\big(kn^8\log^3(m)/\phi^4\big)$ and step size $\eta = O(k/m)$, then with probability at least $1 - O(n^{-1})$, GD is able to find a point that achieves $\epsilon$-training loss within $T = O\big(n^2\log(1/\epsilon)/\phi\big)$ iterations.

For training two-layer ReLU networks, Du et al. [11] made a different assumption on the training data to establish the global convergence of GD. Specifically, Du et al. [11] defined a Gram matrix, which is also known as neural tangent kernel [14], based on the training data $\{\mathbf{x}_i\}_{i=1,\dots,n}$ and assumed that the smallest eigenvalue of such Gram matrix is strictly positive. In fact, for two-layer neural networks, their assumption is equivalent to Assumption 3.2, as shown in the following proposition.

**Proposition 3.6.** Under Assumption 3.1, define the Gram matrix $\mathbf{H} \in \mathbb{R}^{n \times n}$ as follows

$$\mathbf{H}_{ij} = \mathbb{E}_{\mathbf{w} \sim \mathcal{N}(0,\mathbf{I})}[\mathbf{x}_i^\top \mathbf{x}_j \sigma'(\mathbf{w}^\top \mathbf{x}_i) \sigma'(\mathbf{w}^\top \mathbf{x}_j)],$$

then the assumption $\lambda_0 = \lambda_{\min}(\mathbf{H}) > 0$ is equivalent to Assumption 3.2. In addition, there exists a sufficiently small constant $C$ such that $\lambda_0 \geq C\phi n^{-2}$.

**Remark 3.7.** According to Proposition 3.6, we can make a direct comparison between our convergence results for two-layer ReLU networks in Corollary 3.5 with those in Du et al. [11], Oymak and Soltanolkotabi [17]. In specific, as shown in Table 1, the iteration complexity and over-parameterization condition proved in Du et al. [11] can be translated to $O\big(n^6\log(1/\epsilon)/\phi^2\big)$ and $\Omega(n^{14}/\phi^4)$ respectively under Assumption 3.2. Oymak and Soltanolkotabi [17] improved the result in Du et al. [11] and the improved iteration complexity and over-parameterization condition can be translated to $O\big(n^2\|\mathbf{X}\|_2^2\log(1/\epsilon)/\phi\big)$ [2] and $\Omega\big(n^9\|\mathbf{X}\|_2^6/\phi^4\big)$ respectively, where $\mathbf{X} = [\mathbf{x}_1,\dots,\mathbf{x}_n]^\top \in \mathbb{R}^{d\times n}$ is the input data matrix. Our iteration complexity for two-layer ReLU networks is better than that in Oymak and Soltanolkotabi [17] by a factor of $O(\|\mathbf{X}\|_2^2)$ [3], and the over-parameterization condition is also strictly milder than the that in Oymak and Soltanolkotabi [17] by a factor of $O(n\|\mathbf{X}\|_2^6)$.

## 3.2 Extension to training $L$-hidden-layer ReLU networks with SGD

Then we extend the convergence results of GD to SGD in the following theorem.

**Theorem 3.8.** Under Assumptions 3.1 and 3.2, and suppose the number of hidden nodes per layer satisfies

$$m = \Omega\big(kn^{17}L^{12}\log^3(m)/(B^4\phi^8)\big). \tag{3.2}$$

Then if set the step size as $\eta = O\big(kB\phi/(n^3m\log(m))\big)$, with probability at least $1 - O(n^{-1})$, SGD is able to achieve $\epsilon$ expected training loss within

$$T = O\big(n^5L^2\log(m)\log^2(1/\epsilon)/(B\phi^2)\big)$$

iterations.

**Remark 3.9.** We first compare our result with the state-of-the-art proved in Allen-Zhu et al. [2], where they showed that SGD can find a point with $\epsilon$-training loss within $\widetilde{O}\big(n^7 L^2 \log(1/\epsilon)/(B\phi^2)\big)$ iterations if $m = \widetilde{\Omega}\big(n^{24} L^{12} Bk/\phi^8\big)$. In stark contrast, our result on the over-parameterization condition is strictly better than it by a factor of $\widetilde{\Omega}(n^7 B^5)$, and our result on the iteration complexity is also faster by a factor of $O(n^2)$.

Moreover, we also characterize the convergence rate and over-parameterization condition of SGD for training two-layer networks. Unlike the gradient descent, which has the same convergence rate and over-parameterization condition for training both deep and two-layer networks in terms of training data size $n$, we find that the over-parameterization condition of SGD can be further improved for training two-layer neural networks. We state this improved result in the following theorem.

**Theorem 3.10.** Under the same assumptions made in Theorem 3.8. For two-layer ReLU networks, if set the number of hidden nodes and step size as

$$m = \Omega\big(k^{5/2} n^{11} \log^3(m)/(\phi^5 B)\big), \quad \eta = O\big(kB\phi/(n^3 m \log(m))\big),$$

then with probability at least $1 - O(n^{-1})$, stochastic gradient descent is able to achieve $\epsilon$ training loss within $T = O\big(n^5 \log(m) \log(1/\epsilon)/(B\phi^2)\big)$ iterations.

**Remark 3.11.** From Theorem 3.8, we can also obtain the convergence results of SGD for two-layer ReLU networks by choosing $L = 1$. However, the resulting over-parameterization condition is $m = \Omega\big(kn^{17} \log^3(m) B^{-4} \phi^{-8}\big)$, which is much worse than that in Theorem 3.10. This is because for two-layer networks, the training loss enjoys nicer local properties around the initialization, which can be leveraged to improve the convergence of SGD. Due to space limit, we defer more details to Appendix A.

## 4 Proof sketch of the main theory

In this section, we provide the proof sketch for Theorems 3.3, and highlight our technical contributions and innovative proof techniques.

### 4.1 Overview of the technical contributions

The improvements in our result are mainly attributed to the following two aspects: (1) a tighter gradient lower bound leading to faster convergence; and (2) a sharper characterization of the trajectory length of the algorithm.

We first define the following perturbation region based on the initialization,

$$\mathcal{B}(\mathbf{W}^{(0)}, \tau) = \{\mathbf{W} : \|\mathbf{W}_l - \mathbf{W}_l^{(0)}\|_2 \le \tau \text{ for all } l \in [L]\},$$

where $\tau > 0$ is the preset perturbation radius for each weight matrix $\mathbf{W}_l$.

**Tighter gradient lower bound.** By the definition of $\nabla L(\mathbf{W})$, we have $\|\nabla L(\mathbf{W})\|_F^2 = \sum_{l=1}^{L} \|\nabla_{\mathbf{W}_l} L(\mathbf{W})\|_F^2 \ge \|\nabla_{\mathbf{W}_L} L(\mathbf{W})\|_F^2$. Therefore, we can focus on the partial gradient of $L(\mathbf{W})$ with respect to the weight matrix at the last hidden layer. Note that we further have $\|\nabla_{\mathbf{W}_L} L(\mathbf{W})\|_F^2 = \sum_{j=1}^{m} \|\nabla_{\mathbf{w}_{L,j}} L(\mathbf{W})\|_2^2$, where

$$\nabla_{\mathbf{w}_{L,j}} L(\mathbf{W}) = \frac{1}{n} \sum_{i=1}^{n} \langle \mathbf{f}_{\mathbf{W}}(\mathbf{x}_i) - \mathbf{y}_i, \mathbf{v}_j \rangle \sigma'\big(\langle \mathbf{w}_{L,j}, \mathbf{x}_{L-1,i} \rangle\big) \mathbf{x}_{L-1,i},$$

and $\mathbf{x}_{L-1,i}$ denotes the output of the $(L-1)$-th hidden layer with input $\mathbf{x}_i$. In order to prove the gradient lower bound, for each $\mathbf{x}_{L-1,i}$, we introduce a region namely "gradient region", denoted by $\mathcal{W}_j$, which is almost orthogonal to $\mathbf{x}_{L-1,i}$. Then we prove two major properties of these $n$ regions $\{\mathcal{W}_1, \dots, \mathcal{W}_n\}$: (1) $\mathcal{W}_i \cap \mathcal{W}_j = \emptyset$ if $i \ne j$, and (2) if $\mathbf{w}_{L,j} \in \mathcal{W}_i$ for any $i$, with probability at least $1/2$, $\|\nabla_{\mathbf{w}_{L,j}} L(\mathbf{W})\|_2$ is sufficiently large. We visualize these "gradient regions" in Figure 1(a). Since $\{\mathbf{w}_{L,j}\}_{j \in [m]}$ are randomly generated at the initialization, in order to get a larger bound of $\|\nabla_{\mathbf{W}_L} L(\mathbf{W})\|_F^2$, we hope the size of these "gradient regions" to be as large as possible. We take the union of the "gradient regions" for all training data, i.e., $\cup_{i=1}^{n} \mathcal{W}_i$, which is shown in Figure 1(a). As a

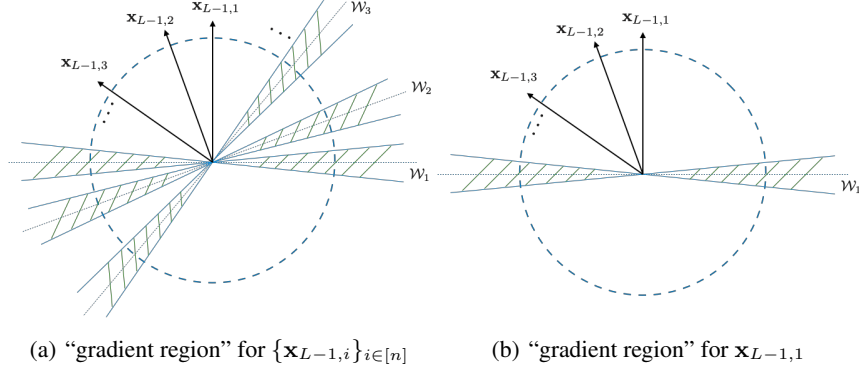

(a) "gradient region" for $\{\mathbf{x}_{L-1,i}\}_{i\in[n]}$        (b) "gradient region" for $\mathbf{x}_{L-1,1}$

Figure 1: (a): "gradient region" for all training data (b): "gradient region" for one training example.

comparison, Allen-Zhu et al. [2], Zou et al. [24] only leveraged the "gradient region" for one training data point to establish the gradient lower bound, which is shown in Figure 1(b). Roughly speaking, the size of "gradient regions" utilized in our proof is $n$ times larger than those used in Allen-Zhu et al. [2], Zou et al. [24], which consequently leads to an $O(n)$ improvement on the gradient lower bound. The improved gradient lower bound is formally stated in the following lemma.

**Lemma 4.1** (Gradient lower bound). Let $\tau = O\big(\phi^{3/2}n^{-3}L^{-6}\log^{-3/2}(m)\big)$, then for all $\mathbf{W} \in \mathcal{B}(\mathbf{W}^{(0)}, \tau)$, with probability at least $1 - \exp\big(O(m\phi/(dn))\big)$, it holds that

$$\|\nabla L(\mathbf{W})\|_F^2 \geq O\big(m\phi L(\mathbf{W})/(kn^2)\big).$$

**Sharper characterization of the trajectory length.** The improved analysis of the trajectory length is motivated by the following observation: at the $t$-th iteration, the decrease of the training loss after one-step gradient descent is proportional to the gradient norm, i.e., $L(\mathbf{W}^{(t)}) - L(\mathbf{W}^{(t+1)}) \propto \|\nabla L(\mathbf{W}^{(t)})\|_F^2$. In addition, the gradient norm $\|\nabla L(\mathbf{W}^{(t)})\|_F$ determines the trajectory length in the $t$-th iteration. Putting them together, we can obtain

$$\|\mathbf{W}_l^{(t+1)} - \mathbf{W}_l^{(t)}\|_2 = \eta\|\nabla_{\mathbf{W}_l}L(\mathbf{W}^{(t)})\|_2 \leq \sqrt{Ckn^2/(m\phi)} \cdot \left(\sqrt{L(\mathbf{W}^{(t)})} - \sqrt{L(\mathbf{W}^{(t+1)})}\right),$$
(4.1)

where $C$ is an absolute constant. (4.1) enables the use of telescope sum, which yields $\|\mathbf{W}_l^{(t)} - \mathbf{W}_l^{(0)}\|_2 \leq \sqrt{Ckn^2L(\mathbf{W}^{(0)})/m\phi}$. In stark contrast, Allen-Zhu et al. [2] bounds the trajectory length as

$$\|\mathbf{W}_l^{(t+1)} - \mathbf{W}_l^{(t)}\|_2 = \eta\|\nabla_{\mathbf{W}_l}L(\mathbf{W}^{(t)})\|_2 \leq \eta\sqrt{C'mL(\mathbf{W}^{(t)})/k},$$

and further prove that $\|\mathbf{W}_l^{(t)} - \mathbf{W}_l^{(0)}\|_2 \leq \sqrt{C'kn^6L^2(\mathbf{W}^{(0)})/(m\phi^2)}$ by taking summation over $t$, where $C'$ is an absolute constant. Our sharp characterization of the trajectory length is formally summarized in the following lemma.

**Lemma 4.2.** Assuming all iterates are staying inside the region $\mathcal{B}(\mathbf{W}^{(0)}, \tau)$ with $\tau = O\big(\phi^{3/2}n^{-3}L^{-6}\log^{-3/2}(m)\big)$, if set the step size $\eta = O\big(k/(L^2m)\big)$, with probability least $1 - O(n^{-1})$, the following holds for all $t \geq 0$ and $l \in [L]$,

$$\|\mathbf{W}_l^{(t)} - \mathbf{W}_l^{(0)}\|_2 \leq O\big(\sqrt{kn^2\log(n)/(m\phi)}\big).$$

## 4.2 Proof of Theorem 3.3

Our proof road map can be organized in three steps: (i) prove that the training loss enjoys good curvature properties within the perturbation region $\mathcal{B}(\mathbf{W}^{(0)}, \tau)$; (ii) show that gradient descent is able to converge to global minima based on such good curvature properties; and (iii) ensure all iterates stay inside the perturbation region until convergence.

**Step (i) Training loss properties.** We first show some key properties of the training loss within $\mathcal{B}(\mathbf{W}^{(0)}, \tau)$, which are essential to establish the convergence guarantees of gradient descent.

**Lemma 4.3.** If $m \geq O(L \log(nL))$, with probability at least $1 - O(n^{-1})$ it holds that $L(\mathbf{W}^{(0)}) \leq \widetilde{O}(1)$.

Lemma 4.3 suggests that the training loss $L(\mathbf{W})$ at the initial point does not depend on the number of hidden nodes per layer, i.e., $m$.

Moreover, the training loss $L(\mathbf{W})$ is nonsmooth due to the non-differentiable ReLU activation function. Generally speaking, smoothness is essential to achieve linear rate of convergence for gradient-based algorithms. Fortunately, Allen-Zhu et al. [2] showed that the training loss satisfies locally semi-smoothness property, which is summarized in the following lemma.

**Lemma 4.4** (Semi-smoothness [2])**.** Let

$$\tau \in \left[\Omega\big(k^{3/2}/(m^{3/2}L^{3/2}\log^{3/2}(m))\big), O\big(1/(L^{4.5}\log^{3/2}(m))\big)\right].$$

Then for any two collections $\widehat{\mathbf{W}} = \{\widehat{\mathbf{W}}_l\}_{l \in [L]}$ and $\widetilde{\mathbf{W}} = \{\widetilde{\mathbf{W}}_l\}_{l \in [L]}$ satisfying $\widehat{\mathbf{W}}, \widetilde{\mathbf{W}} \in \mathcal{B}(\mathbf{W}^{(0)}, \tau)$, with probability at least $1 - \exp(-\Omega(-m\tau^{3/2}L))$, there exist two constants $C'$ and $C''$ such that

$$L(\widetilde{\mathbf{W}}) \leq L(\widehat{\mathbf{W}}) + \langle \nabla L(\widehat{\mathbf{W}}), \widetilde{\mathbf{W}} - \widehat{\mathbf{W}} \rangle$$
$$+ C'\sqrt{L(\widehat{\mathbf{W}})} \cdot \frac{\tau^{1/3}L^2\sqrt{m\log(m)}}{\sqrt{k}} \cdot \|\widetilde{\mathbf{W}} - \widehat{\mathbf{W}}\|_2 + \frac{C''L^2m}{k}\|\widetilde{\mathbf{W}} - \widehat{\mathbf{W}}\|_2^2. \quad (4.2)$$

Lemma 4.4 is a rescaled version of Theorem 4 in Allen-Zhu et al. [2], since the training loss $L(\mathbf{W})$ in (2.1) is divided by the training sample size $n$, as opposed to the training loss in Allen-Zhu et al. [2]. This lemma suggests that if the perturbation region is small, i.e., $\tau \ll 1$, the non-smooth term (third term on the R.H.S. of (4.2)) is small and dominated by the gradient term (the second term on the the R.H.S. of (4.2)). Therefore, the training loss behaves like a smooth function in the perturbation region and the linear rate of convergence can be proved.

**Step (ii) Convergence rate of GD.** Now we are going to establish the convergence rate for gradient descent under the assumption that all iterates stay inside the region $\mathcal{B}(\mathbf{W}^{(0)}, \tau)$, where $\tau$ will be specified later.

**Lemma 4.5.** Assume all iterates stay inside the region $\mathcal{B}(\mathbf{W}^{(0)}, \tau)$, where $\tau = O\big(\phi^{3/2}n^{-3}L^{-6}\log^{-3/2}(m)\big)$. Then under Assumptions 3.1 and 3.2, if set the step size $\eta = O\big(k/(L^2m)\big)$, with probability least $1 - \exp\big(-O(m\tau^{3/2}L)\big)$, it holds that

$$L(\mathbf{W}^{(t)}) \leq \left(1 - O\left(\frac{m\phi\eta}{kn^2}\right)\right)^t L(\mathbf{W}^{(0)}).$$

Lemma 4.5 suggests that gradient descent is able to decrease the training loss to zero at a linear rate.

**Step (iii) Verifying all iterates of GD stay inside the perturbation region.** Then we are going to ensure that all iterates of GD are staying inside the required region $\mathcal{B}(\mathbf{W}^{(0)}, \tau)$. Note that we have proved the distance $\|\mathbf{W}_l^{(t)} - \mathbf{W}_l^{(0)}\|_2$ in Lemma 4.2. Therefore, it suffices to verify that such distance is smaller than the preset value $\tau$. Thus, we can complete the proof of Theorem 3.3 by verifying the conditions based on our choice of $m$. Note that we have set the required number of $m$ in (3.1), plugging (3.1) into the result of Lemma 4.2, we have with probability at least $1 - O(n^{-1})$, the following holds for all $t \leq T$ and $l \in [L]$

$$\|\mathbf{W}_l^{(t)} - \mathbf{W}_l^{(0)}\|_2 \leq O\big(\phi^{3/2}n^{-3}L^{-6}\log^{-3/2}(m)\big),$$

which is exactly in the same order of $\tau$ in Lemma 4.5. Therefore, our choice of $m$ guarantees that all iterates are inside the required perturbation region. In addition, by Lemma 4.5, in order to achieve $\epsilon$ accuracy, we require

$$T\eta = O\big(kn^2\log\big(1/\epsilon\big)m^{-1}\phi^{-1}\big). \quad (4.3)$$

Then substituting our choice of step size $\eta = O\big(k/(L^2m)\big)$ into (4.3) and applying Lemma 4.3, we can get the desired result for $T$.

### 4.3 Optimizing both top and hidden layers

Here we would like to briefly discuss the extension to the case where the top layer is also optimized. The proof sketch is as follows: similar to our current proof, we can also define a small perturbation region around the initialization, but the new definition involves a constraint on the top layer weights. Specifically, such new perturbation region can be defined as follows,

$$\mathcal{B}(\mathbf{W}^{(0)}, \tau) = \{\mathbf{W} : \|\mathbf{W}_l - \mathbf{W}_l^{(0)}\|_2 \leq \tau \text{ for all } l \in [L], \ \|\mathbf{V} - \mathbf{V}^{(0)}\|_2 \leq \tau'\}.$$

Then, it can be proved that the neural network also enjoys good properties inside such region. Similar to the proof in this paper, based on these good properties, we can prove that until convergence the neural network weights, including the top layer weights, would not escape from such region. Note that optimizing more parameter can lead to larger gradient, thus we can prove a larger gradient lower bound during the training process which can potential speed up the convergence of optimization algorithm (e.g., GD, SGD).

## 5 Conclusions and future work

In this paper, we studied the global convergence of (stochastic) gradient descent for training over-parameterized ReLU networks, and improved the state-of-the-art results. Our proof technique can be also applied to prove similar results for other loss functions such as cross-entropy loss and other neural network architectures such as convolutional neural networks (CNN) [2, 11] and ResNet [2, 11, 21]. One important future work is to investigate whether the over-parameterization condition and the convergence rate can be further improved. It is promising that if we can further improve the characterization of "gradient region", as it may provide a tighter gradient lower bound and consequently sharpen the over-parameterization condition. Another interesting future direction is to explore the use of our proof technique to improve the generalization analysis of overparameterized neural networks trained by gradient-based algorithms [1, 6, 4].

## Acknowledgement

We thank the anonymous reviewers and area chair for their helpful comments. We also thank Jinshan Zeng for his helpful comment on the proof in the earlier version of our work. This research was sponsored in part by the National Science Foundation CAREER Award IIS-1906169, BIGDATA IIS-1855099, and Salesforce Deep Learning Research Award. The views and conclusions contained in this paper are those of the authors and should not be interpreted as representing any funding agencies.

## Footnotes

[1]Here $\widetilde{\Omega}(\cdot)$ hides constants and the logarithmic dependencies on problem dependent parameters except $\epsilon$.

[2]It is worth noting that $\|\mathbf{X}\|_2^2 = O(1)$ if $d \lesssim n$, $\|\mathbf{X}\|_2^2 = O(n/d)$ if $\mathbf{X}$ is randomly generated, and $\|\mathbf{X}\|_2^2 = O(n)$ in the worst case.

[3]Here we set $k = 1$ in order to match the problem setting in Du et al. [11], Oymak and Soltanolkotabi [17].

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
