[Supplementary Material · deep_relu_improved (6).pdf]

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

# A Proof of the Main Theory

## A.1 Proof of Proposition 3.6

We prove this proposition by two steps: (1) we prove that if there is no duplicate training data, it must hold that $\lambda_{\min}(\mathbf{H}) > 0$; (2) we prove that if there exists at least one duplicate training data, we have $\lambda_{\min}(\mathbf{H}) = 0$.

The first step can be done by applying Theorem 3 in Du et al. [11], where the author showed that if for any $i \neq j$, $\mathbf{x}_i \nparallel \mathbf{x}_j$, then it holds that $\lambda_{\min}(\mathbf{H}) > 0$. Since under Assumption 3.1, we have $\|\mathbf{x}_i\|_2 = \|\mathbf{x}_j\|_2$. Then it can be shown that $\mathbf{x}_i \neq \mathbf{x}_j$ for all $i \neq j$ is an sufficient condition to $\lambda_{\min}(\mathbf{H})$.

Then we conduct the second step. Clearly, if we have two training data with $\mathbf{x}_i = \mathbf{x}_j$, it can be shown that $\mathbf{H}_{ik} = \mathbf{H}_{jk}$ for all $k = 1, \ldots, n$. This immediately implies that there exist two identical rows in $\mathbf{H}$, which further suggests that $\lambda_{\min}(\mathbf{H}) = 0$.

The last argument can be directly proved by Lemma I.1 in Oymak and Soltanolkotabi [17], where the authors showed that $\lambda_0 = \lambda_{\min}(\mathbf{H}) \geq \phi/(100n^2)$.

By combining the above discussions, we are able to complete the proof.

## A.2 Proof of Theorem 3.8

Now we sketch the proof of Theorem 3.8. Following the same idea of proving Theorem 3.3, we split the whole proof into three steps.

**Step (i) Initialization and perturbation region characterization.** Unlike the proof for GD, in addition to the crucial gradient lower bound specified in Lemma 4.1, we also require the gradient upper bound, which is stated in the following lemma.

**Lemma A.1** (Gradient upper bounds [2]). Let $\tau = O\big(\phi^{3/2}n^{-3}L^{-6}\log^{-3/2}(m)\big)$, then for all $\mathbf{W} \in \mathcal{B}(\mathbf{W}^{(0)}, \tau)$, with probability at least $1 - \exp\big(O(m\phi/(dn))\big)$, the following holds for all $l \in [L]$

$$\|\nabla_{\mathbf{W}_l}L(\mathbf{W})\|_F^2 \leq O\left(\frac{mL(\mathbf{W})}{k}\right), \quad \|\nabla_{\mathbf{W}_l}\ell(\mathbf{f}_{\mathbf{W}}(\mathbf{x}_i), \mathbf{y}_i)\|_F^2 \leq O\left(\frac{m\ell(\mathbf{f}_{\mathbf{W}}(\mathbf{x}_i), \mathbf{y}_i)}{k}\right).$$

In later analysis, we show that the gradient upper bound will be exploited to bound the distance between iterates of SGD and its initialization. Besides, note that Lemmas 4.3 and 4.4 hold for both GD and SGD, we do not state them again in this part.

**Step (ii) Convergence rate of SGD.** Analogous to the proof for GD, the following lemma shows that SGD is able to converge to the global minima at a linear rate.

**Lemma A.2.** Assume all iterates stay inside the region $\mathcal{B}(\mathbf{W}^{(0)}, \tau)$, where $\tau = O\big(\phi^3 B^{3/2}n^{-6}L^{-6}\log^{-3/2}(m)\big)$. Then under Assumptions 3.1 and 3.2, if set the step size $\eta = O\big(B\phi/(L^2mn^2)\big)$, with probability least $1 - \exp\big(-O(m\tau^{3/2}L)\big)$, it holds that

$$\mathbb{E}[L(\mathbf{W}^{(t)})] \leq \left(1 - O\left(\frac{m\phi\eta}{kn^2}\right)\right)^t L(\mathbf{W}^{(0)}).$$

**Step (iii) Verifying all iterates of SGD stay inside the perturbation region.** Similar to the proof for GD, the following lemma characterizes the distance from each iterate to the initial point for SGD.

**Lemma A.3.** Under the same assumptions made in Lemma A.2, if set the step size $\eta = O\big(kB\phi/(n^3L^2m\log(m))\big)$, suppose $m \geq O(T \cdot n)$, with probability at least $1 - O(n^{-1})$, the following holds for all $t \leq T$ and $l \in [L]$,

$$\|\mathbf{W}_l^{(t)} - \mathbf{W}_l^{(0)}\|_2 \leq O\big(k^{1/2}n^{5/2}B^{-1/2}m^{-1/2}\phi^{-1}\big).$$

*Proof of Theorem 3.8.* Compared with Lemma 4.2, the trajectory length of SGD is much larger than that of GD. In addition, we require a much smaller step size to guarantee that the iterates do not go

too far away from the initial point. This makes over-parameterization condition of SGD worse than that of GD.

We complete the proof of Theorem 3.8 by verifying our choice of $m$ in (3.2). By substituting (3.2) into Lemma A.3, we have with probability at least $1 - O(n^{-1})$, the following holds for all $t \leq T$ and $l \in [L]$

$$\|\mathbf{W}_l^{(t)} - \mathbf{W}_l^{(0)}\|_2 = O\big(\phi^{3/2} B^{3/2} n^{-6} L^{-6} \log^{-3/2}(m)\big),$$

which is exactly in the same order of $\tau$ in Lemma A.2. Then by Lemma A.2, we know that in order to achieve $\epsilon$ expected training loss, it suffices to set

$$T\eta = O\big(kn^2 m^{-1} \phi^{-1} \log(1/\epsilon)\big).$$

Then applying our choice of step size, i.e., $\eta = O\big(kB\phi/(n^3 L^2 m \log(m))\big)$, we can get the desired result for $T$. This completes the proof. $\qquad\square$

## A.3 Proof of Theorem 3.10

Before proving Theorem 3.10, we first deliver the following two lemmas. The first lemma states the upper bound of stochastic gradient in $\|\cdot\|_{2,\infty}$ norm.

**Lemma A.4.** With probability at least $1 - O(m^{-1})$, it holds that

$$\|\nabla\ell(\mathbf{f}_{\mathbf{W}}(\mathbf{x}_i), \mathbf{y}_i)\|_{2,\infty}^2 \leq O\big(\ell(\mathbf{f}_{\mathbf{W}}(\mathbf{x}_i), \mathbf{y}_i) \cdot \log(m)\big)$$

for all $\mathbf{W} \in \mathbb{R}^{m \times d}$ and $i \in [n]$.

The following lemma gives a different version of semi-smoothness for two-layer ReLU network.

**Lemma A.5** (Semi-smoothness for two-layer ReLU network)**.** For any two collections $\widehat{\mathbf{W}} = \{\widehat{\mathbf{W}}_l\}_{l \in [L]}$ and $\widetilde{\mathbf{W}} = \{\widetilde{\mathbf{W}}_l\}_{l \in [L]}$ satisfying $\widehat{\mathbf{W}}, \widetilde{\mathbf{W}} \in \mathcal{B}(\mathbf{W}^{(0)}, \tau)$, with probability at least $1 - \exp(-O(-m\tau^{2/3}))$, there exist two constants $C'$ and $C''$ such that

$$L(\widetilde{\mathbf{W}}) \leq L(\widehat{\mathbf{W}}) + \langle \nabla L(\widehat{\mathbf{W}}), \widetilde{\mathbf{W}} - \widehat{\mathbf{W}} \rangle$$
$$+ C'\sqrt{L(\widehat{\mathbf{W}})} \cdot \frac{\tau^{2/3} m \sqrt{\log(m)}}{\sqrt{k}} \cdot \|\widetilde{\mathbf{W}} - \widehat{\mathbf{W}}\|_{2,\infty} + \frac{C''m}{k} \|\widetilde{\mathbf{W}} - \widehat{\mathbf{W}}\|_2^2.$$

It is worth noting that Lemma 4.4 can also imply a $\|\cdot\|_{2,\infty}$ norm based semi-smoothness result by applying the inequality $\|\widetilde{\mathbf{W}} - \widehat{\mathbf{W}}\|_2 \leq \|\widetilde{\mathbf{W}} - \widehat{\mathbf{W}}\|_F \leq \sqrt{m}\|\widetilde{\mathbf{W}} - \widehat{\mathbf{W}}\|_{2,\infty}$. However, this operation will maintain the dependency on $\tau$, i.e., $\tau^{1/3}$, which is worse than that in Lemma A.5 (e.g. $\tau^{2/3}$) since typically we have $\tau \ll 1$. Therefore, Lemma A.5 is crucial to establish a better convergence guarantee for SGD in training two-layer ReLU network.

*Proof of Theorem 3.10.* To simplify the proof, we use the following short-hand notation to define mini-batch stochastic gradient at the $t$-th iteration

$$\mathbf{G}^{(t)} = \frac{1}{|\mathcal{B}^{(t)}|} \sum_{s \in \mathcal{B}^{(t)}} \nabla\ell\big(\mathbf{f}_{\mathbf{W}^{(t)}}(\mathbf{x}_s), \mathbf{y}_s\big),$$

where $\mathcal{B}^{(t)}$ is the minibatch of data indices with $|\mathcal{B}^{(t)}| = B$. Then we bound its variance as follows,

$$\mathbb{E}[\|\mathbf{G}^{(t)} - \nabla L(\mathbf{W}^{(t)})\|_F^2] \leq \frac{1}{B}\mathbb{E}_s[\|\nabla\ell\big(\mathbf{f}_{\mathbf{W}^{(t)}}(\mathbf{x}_s), \mathbf{y}_s\big) - \nabla L(\mathbf{W}^{(t)})\|_F^2]$$
$$\leq \frac{2}{B}\big[\mathbb{E}_s[\|\nabla\ell\big(\mathbf{f}_{\mathbf{W}^{(t)}}(\mathbf{x}_s), \mathbf{y}_s\big)\|_F^2] + \|\nabla L(\mathbf{W}^{(t)})\|_F^2\big]$$
$$\leq \frac{4CL(\mathbf{W}^{(t)})}{Bk}, \tag{A.1}$$

where $C$ is an absolute constant, the expectation is taken over the random choice of training data and the second inequality follows from Young's inequality and the last inequality is by Lemma A.1. Moreover, we can further bound the expectation $\mathbb{E}[\|\mathbf{G}^{(t)}\|_2^2]$ as follows,

$$\mathbb{E}[\|\mathbf{G}^{(t)}\|_2^2] \leq 2\mathbb{E}[\|\mathbf{G}^{(t)} - \nabla L(\mathbf{W}^{(t)})\|_F^2] + 2\|\nabla L(\mathbf{W}^{(t)})\|_F^2 \leq \frac{8CmL(\mathbf{W}^{(t)})}{Bk} + 2\|\nabla L(\mathbf{W}^{(t)})\|_F^2. \tag{A.2}$$

By Lemma A.5, we have the following for one-step stochastic gradient descent

$$L(\mathbf{W}^{(t+1)}) \leq L(\mathbf{W}^{(t)}) - \eta\langle\nabla L(\mathbf{W}^{(t)}), \mathbf{G}^{(t)}\rangle$$
$$+ C'\eta\sqrt{L(\mathbf{W}^{(t)})} \cdot \frac{\tau^{2/3}m\sqrt{\log(m)}}{\sqrt{k}} \cdot \|\mathbf{G}^{(t)}\|_{2,\infty} + \frac{C''m\eta^2}{k} \cdot \|\mathbf{G}^{(t)}\|_2^2.$$

Taking expectation conditioned on $\mathbf{W}^{(t)}$, we obtain

$$\mathbb{E}[L(\mathbf{W}^{(t+1)})|\mathbf{W}^{(t)}] \leq L(\mathbf{W}^{(t)}) - \eta\langle\nabla L(\mathbf{W}^{(t)}), \nabla L(\mathbf{W}^{(t)})\rangle$$
$$+ C'\eta\sqrt{L(\mathbf{W}^{(t)})} \cdot \frac{\tau^{2/3}m\sqrt{\log(m)}}{\sqrt{k}} \cdot \mathbb{E}[\|\mathbf{G}^{(t)}\|_{2,\infty}|\mathbf{W}^{(t)}]$$
$$+ \frac{C''m\eta^2}{k} \cdot \mathbb{E}[\|\mathbf{G}^{(t)}\|_2^2|\mathbf{W}^{(t)}]. \tag{A.3}$$

By Lemma A.4, with probability at least $1 - O(m^{-1})$ we have the following upper bound on the quantity $\mathbb{E}[\|\mathbf{G}^{(t)}\|_{2,\infty}|\mathbf{W}^{(t)}]$ for all $t = 1, \ldots, T$,

$$\mathbb{E}[\|\mathbf{G}^{(t)}\|_{2,\infty}|\mathbf{W}^{(t)}] \leq \mathbb{E}[\|\nabla\ell(\mathbf{f}_{\mathbf{W}^{(t)}}(\mathbf{x}_i), \mathbf{y}_i)\|_{2,\infty}|\mathbf{W}^{(t)}] \leq O\big(\sqrt{L(\mathbf{W}^{(t)})\log(m)}\big).$$

Then based on Lemma 4.1, plugging (A.2) and the above inequality into (A.3), and set

$$\eta = O\left(\frac{k}{mn^2}\right) \quad \text{and} \quad \tau = O\left(\frac{\phi^3}{n^3k^{3/4}\log^{3/2}(m)}\right).$$

Then with proper adjustment of constants we can obtain

$$\mathbb{E}[L(\mathbf{W}^{(t+1)})|\mathbf{W}^{(t)}] \leq L(\mathbf{W}^{(t)}) - \frac{\eta}{2}\|\nabla L(\mathbf{W}^{(t)})\|_F^2 \leq \left(1 - \frac{m\phi\eta}{2kn^2}\right)L(\mathbf{W}^{(t)}),$$

where the last inequality follows from Lemma 4.1. Then taking expectation on $\mathbf{W}^{(t)}$, we have with probability $1 - O(m^{-1})$,

$$\mathbb{E}[L(\mathbf{W}^{(t+1)})] \leq \left(1 - \frac{m\phi\eta}{2kn^2}\right)\mathbb{E}[L(\mathbf{W}^{(t)})] \leq \left(1 - \frac{m\phi\eta}{2kn^2}\right)^{t+1}\mathbb{E}[L(\mathbf{W}^{(0)})] \tag{A.4}$$

holds for all $t > 0$. Then by Lemma A.3, we know that if set $\eta = O\big(kB\phi/(n^3m\log(m))\big)$, with probability at least $1 - O(n^{-1})$, it holds that

$$\|\mathbf{W}_l^{(t)} - \mathbf{W}_l^{(0)}\|_2 \leq O\left(\frac{k^{1/2}n^{5/2}}{B^{1/2}m^{1/2}\phi}\right),$$

for all $t \leq T$. Then by our choice of $m$, it is easy to verify that with probability at least $1 - O(n^{-1}) - O(m^{-1}) = 1 - O(n^{-1})$,

$$\|\mathbf{W}_l^{(t)} - \mathbf{W}_l^{(0)}\|_2 \leq O\left(\frac{k^{1/2}n^{5/2}}{B^{1/2}\phi} \cdot \frac{\phi^4 B^{1/2}}{k^{5/4}n^{11/2}\log^{3/2}(m)}\right) = \tau.$$

Moreover, note that in Lemma A.3 we set the step size as $\eta = O\big(kB\phi/(n^3m\log(m))\big)$ and (A.4) suggests that we need

$$T\eta = O\left(\frac{kn^2}{m\phi}\right)$$

to achieve $\epsilon$ expected training loss. Therefore we can derive the number of iteration as

$$T = O\left(\frac{n^5\log(m)\log(1/\epsilon)}{B\phi^2}\right).$$

This completes the proof. □

# B Proof of Lemmas in Section 4 and Appendix A

## B.1 Proof of Lemma 4.1

We first provide the following useful lemmas before starting the proof of Lemma 4.1.

The following lemma states that with high probability the norm of the output of each hidden layer is bounded by constants.

**Lemma B.1** ([24]). At the initialization, if $m \geq O(L \log(nL))$, with probability at least $1 - \exp(-O(m/L))$, it holds that $1/2 \leq \|\mathbf{x}_{l,i}\|_2 \leq 2$ and $\big\|\mathbf{x}_{l,i}/\|\mathbf{x}_{l,i}\|_2 - \mathbf{x}_{l,j}/\|\mathbf{x}_{l,j}\|_2\big\|_2 \geq \phi/2$ for all $i, j \in [n]$ and $l \in [L]$, where $\mathbf{x}_{l,i}$ denotes the output of the $l$-th hidden layer given the input $\mathbf{x}_i$ and initial weight matrices $\mathbf{W}^{(0)}$.

The following lemma states

**Lemma B.2.** Assume $m \geq \widetilde{O}(n^2 k^2 \phi^{-1})$, then there exist an absolute constant $C > 0$ such that with probability at least $1 - \exp\big(-O(m\phi/(kn))\big)$, it holds that

$$\sum_{j=1}^{m} \left\| \frac{1}{n} \sum_{i=1}^{n} \langle \mathbf{u}_i, \mathbf{v}_j \rangle \sigma'(\langle \mathbf{w}_{L,j}^{(0)}, \mathbf{x}_{L-1,i} \rangle) \mathbf{x}_{L-1,i} \right\|_2^2 \geq \frac{C\phi m \sum_{i=1}^{n} \|\mathbf{u}_i\|_2^2}{kn^3}.$$

If we set $\mathbf{u}_i = \mathbf{f}_{\mathbf{W}^{(0)}}(\mathbf{x}_i) - \mathbf{y}_i$, Lemma B.2 corresponds to the gradient lower bound at the initialization. Then the next step is to prove the bounds for all $\mathbf{W}$ in the required perturbation region. Before proceeding to our final proof, we present the following lemma that provides useful results regarding the neural network within the perturbation region.

**Lemma B.3** ([2]). Consider a collection of weight matrices $\widetilde{\mathbf{W}} = \{\widetilde{\mathbf{W}}_l\}_{l=1,\dots,L}$ such that $\widetilde{\mathbf{W}} \in \mathcal{B}(\mathbf{W}^{(0)}, \tau)$, with probability at least $1 - \exp(-O(m\tau^{2/3}L))$, there exists constants $C'$, $C''$ and $C'''$ such that

- $\left\|\widetilde{\mathbf{\Sigma}}_{L,i} - \mathbf{\Sigma}_{L,i}\right\|_0 \leq C'\tau^{2/3}L$

- $\|\mathbf{V}(\widetilde{\mathbf{\Sigma}}_{L,i} - \mathbf{\Sigma}_{L,i})\|_2 \leq C''\tau^{1/3}L^2\sqrt{m\log(m)}/\sqrt{k}$

- $\|\widetilde{\mathbf{x}}_{L-1,i} - \mathbf{x}_{L-1,i}\|_2 \leq C'''\tau L^{5/2}\sqrt{\log(m)}$,

for all $i = 1, \dots, n$, where $\mathbf{x}_{L-1,i}$ and $\widetilde{\mathbf{x}}_{L-1,i}$ denote the outputs of the $L-1$-th layer of the neural network with weight matrices $\mathbf{W}^{(0)}$ and $\widetilde{\mathbf{W}}$, and $\mathbf{\Sigma}_{L,i}$ and $\widetilde{\mathbf{\Sigma}}_{L,i}$ are diagonal matrices with $(\mathbf{\Sigma}_{L,i})_{jj} = \sigma'(\langle \mathbf{w}_{L,j}^{(0)}, \mathbf{x}_{L-1} \rangle)$ and $(\widetilde{\mathbf{\Sigma}}_{L,i})_{jj} = \sigma'(\langle \widetilde{\mathbf{w}}_{L,j}, \widetilde{\mathbf{x}}_{L-1} \rangle)$ respectively.

Now we are ready to prove the lower and upper bounds of the Frobenious norm of the gradient.

*Proof of Lemma 4.1.* Consider any $\widetilde{\mathbf{W}} \in \mathcal{B}(\mathbf{W}^{(0)}, \tau)$, the gradient $\nabla_{\mathbf{W}_L} L(\widetilde{\mathbf{W}})$ takes form

$$\nabla_{\mathbf{W}_L} L(\widetilde{\mathbf{W}}) = \frac{1}{n} \sum_{i=1}^{n} \left( (\mathbf{f}_{\widetilde{\mathbf{W}}}(\mathbf{x}_i) - \mathbf{y}_i)^\top \mathbf{V} \widetilde{\mathbf{\Sigma}}_{L,i} \right)^\top \widetilde{\mathbf{x}}_{L-1,i}^\top,$$

where $\widetilde{\mathbf{\Sigma}}_{L,i}$ is a diagonal matrix with $(\widetilde{\mathbf{\Sigma}}_{L,i})_{jj} = \sigma'(\widetilde{\mathbf{w}}_{L-1,j}, \widetilde{\mathbf{x}}_{L-1,i})$ and $\widetilde{\mathbf{x}}_{l-1,i}$ denotes the output of the $l$-th hidden layer with input $\mathbf{x}_i$ and model weight matrices $\widetilde{\mathbf{W}}$. Let $\mathbf{v}_j^\top$ denote the $j$-th row of matrix $\mathbf{V}$, and define

$$\widetilde{\mathbf{G}} = \frac{1}{n} \sum_{i=1}^{n} \left( (\mathbf{f}_{\widetilde{\mathbf{W}}}(\mathbf{x}_i) - \mathbf{y}_i)^\top \mathbf{V} \mathbf{\Sigma}_{L,i} \right)^\top \mathbf{x}_{L-1,i}^\top,$$

where $\mathbf{\Sigma}_{L,i}$ is a diagonal matrix with $(\mathbf{\Sigma}_{L,i})_{jj} = \sigma'(\mathbf{w}_{L-1,j}^{(0)}, \mathbf{x}_{L-1,i})$. Then by Lemma B.2, we have with probability at least $1 - \exp(-O(m\phi/(kn)))$, the following holds for any $\widetilde{\mathbf{W}}$,

$$
\begin{aligned}
\|\widetilde{\mathbf{G}}\|_F^2 &= \frac{1}{n^2} \sum_{j=1}^m \left\| \sum_{i=1}^n \langle \mathbf{f}_{\widetilde{\mathbf{W}}}(\mathbf{x}_i) - \mathbf{y}_i, \mathbf{v}_j \rangle \sigma'(\langle \mathbf{w}_{L,j}, \mathbf{x}_{L-1,i} \rangle \mathbf{x}_{L-1,i}) \right\|_2^2 \\
&\geq \frac{C_0 \phi m \sum_{i=1}^n \|\mathbf{f}_{\widetilde{\mathbf{W}}}(\mathbf{x}_i) - \mathbf{y}_i\|_2^2}{kn^3},
\end{aligned}
$$

where $C_0$ is an absolute constant. Then we have

$$
\begin{aligned}
&\left\| \widetilde{\mathbf{G}} - \nabla_{\mathbf{W}_L} L(\widetilde{\mathbf{W}}) \right\|_F \\
&= \frac{1}{n} \left\| \sum_{i=1}^n \left( (\mathbf{f}_{\widetilde{\mathbf{W}}}(\mathbf{x}_i) - \mathbf{y}_i)^\top \mathbf{V} \mathbf{\Sigma}_{L,i} \right)^\top \mathbf{x}_{L-1,i}^\top - \sum_{i=1}^n \left( (\mathbf{f}_{\widetilde{\mathbf{W}}}(\mathbf{x}_i) - \mathbf{y}_i)^\top \mathbf{V} \widetilde{\mathbf{\Sigma}}_{L,i} \right)^\top \widetilde{\mathbf{x}}_{L-1,i}^\top \right\|_F \\
&\leq \frac{1}{n} \Bigg[ \left\| \sum_{i=1}^n \left( (\mathbf{f}_{\widetilde{\mathbf{W}}}(\mathbf{x}_i) - \mathbf{y}_i)^\top \mathbf{V} (\mathbf{\Sigma}_{L,i} - \widetilde{\mathbf{\Sigma}}_{L,i}) \right)^\top \mathbf{x}_{L-1,i}^\top \right\|_F \\
&\quad + \left\| \sum_{i=1}^n \left( (\mathbf{f}_{\widetilde{\mathbf{W}}}(\mathbf{x}_i) - \mathbf{y}_i)^\top \mathbf{V} \widetilde{\mathbf{\Sigma}}_{L,i} \right)^\top \left( \mathbf{x}_{L-1,i} - \widetilde{\mathbf{x}}_{L-1,i} \right)^\top \right\|_F \Bigg].
\end{aligned}
$$

By Lemmas B.1 and B.3, we have

$$
\begin{aligned}
&\left\| \sum_{i=1}^n \left( (\mathbf{f}_{\widetilde{\mathbf{W}}}(\mathbf{x}_i) - \mathbf{y}_i)^\top \mathbf{V} (\mathbf{\Sigma}_{L,i} - \widetilde{\mathbf{\Sigma}}_{L,i}) \right)^\top \mathbf{x}_{L-1,i}^\top \right\|_F \\
&\leq \sum_{i=1}^n \left\| \mathbf{f}_{\widetilde{\mathbf{W}}}(\mathbf{x}_i) - \mathbf{y}_i \right\|_2 \left\| \mathbf{V} (\mathbf{\Sigma}_{L,i} - \widetilde{\mathbf{\Sigma}}_{L,i}) \right\|_2 \| \mathbf{x}_{L-1,i} \|_2 \\
&\leq \frac{C_1 \tau^{1/3} L^2 \sqrt{m \log(m)}}{\sqrt{k}} \cdot \sum_{i=1}^n \left\| \mathbf{f}_{\widetilde{\mathbf{W}}(\mathbf{x}_i)} - \mathbf{y}_i \right\|_2,
\end{aligned}
$$

where the second inequality follows from Lemma B.3 and $C_1$ is an absolute constant. In addition, we also have

$$
\begin{aligned}
&\left\| \sum_{i=1}^n \left( (\mathbf{f}_{\widetilde{\mathbf{W}}}(\mathbf{x}_i) - \mathbf{y}_i)^\top \mathbf{V} \widetilde{\mathbf{\Sigma}}_{L,i} \right)^\top \left( \mathbf{x}_{L-1,i} - \widetilde{\mathbf{x}}_{L-1,i} \right)^\top \right\|_F \\
&\leq \sum_{i=1}^n \| \mathbf{f}_{\widetilde{\mathbf{W}}}(\mathbf{x}_i) - \mathbf{y}_i \|_2 \| \mathbf{V} \|_2 \| \mathbf{x}_{L-1,i} - \widetilde{\mathbf{x}}_{L-1,i} \|_2 \\
&\leq \frac{C_2 \tau L^{5/2} \sqrt{m \log(m)}}{\sqrt{k}} \cdot \sum_{i=1}^n \| \mathbf{f}_{\widetilde{\mathbf{W}}}(\mathbf{x}_i) - \mathbf{y}_i \|_2,
\end{aligned}
$$

where the second inequality follows from Lemma B.3 and $C_2$ is an absolute constant. Combining the above bounds we have

$$
\begin{aligned}
\left\| \widetilde{\mathbf{G}} - \nabla_{\mathbf{W}_L} L(\widetilde{\mathbf{W}}) \right\|_F &\leq \frac{\sum_{i=1}^n \| \mathbf{f}_{\widetilde{\mathbf{W}}}(\mathbf{x}_i) - \mathbf{y}_i \|_2}{n} \cdot \left( \frac{C_1 \tau^{1/3} L^2 \sqrt{m \log(m)}}{\sqrt{k}} + \frac{C_2 \tau L^{5/2} \sqrt{m \log(m)}}{\sqrt{k}} \right) \\
&\leq \frac{\sum_{i=1}^n \| \mathbf{f}_{\widetilde{\mathbf{W}}}(\mathbf{x}_i) - \mathbf{y}_i \|_2}{n} \cdot \frac{C_3 \tau^{1/3} L^2 \sqrt{m \log(m)}}{\sqrt{k}},
\end{aligned}
$$

where the second inequality follows from the fact that $\tau \leq O(L^{-4/3})$. Then by triangle inequality, we have the following lower bound of $\|\nabla_{\mathbf{W}_L} L(\widetilde{\mathbf{W}})\|_F$

$$
\begin{aligned}
\|\nabla_{\mathbf{W}_L} L(\widetilde{\mathbf{W}})\|_F &\geq \|\widetilde{\mathbf{G}}\|_F - \|\widetilde{\mathbf{G}} - \nabla_{\mathbf{W}_L} L(\widetilde{\mathbf{W}})\|_F \\
&\geq \frac{C_0 \phi^{1/2} m^{1/2} \sqrt{n \sum_{i=1}^n \|\mathbf{f}_{\widetilde{\mathbf{W}}}(\mathbf{x}_i) - \mathbf{y}_i\|_2^2}}{\sqrt{k} n^2} \\
&\quad - \frac{\sum_{i=1}^n \|\mathbf{f}_{\widetilde{\mathbf{W}}}(\mathbf{x}_i) - \mathbf{y}_i\|_2}{n} \cdot \frac{C_3 \tau^{1/3} L^2 \sqrt{m \log(m)}}{\sqrt{k}}.
\end{aligned}
$$

By Jensen's inequality we know that $n \sum_{i=1}^{n} \|\mathbf{f}_{\widetilde{\mathbf{W}}}(\mathbf{x}_i) - \mathbf{y}_i\|_2^2 \geq \left( \sum_{i=1}^{n} \|\mathbf{f}_{\widetilde{\mathbf{W}}}(\mathbf{x}_i) - \mathbf{y}_i\|_2 \right)^2$. Then we set

$$\tau = \frac{C_3 \phi^{3/2}}{2 C_0 n^3 L^6 \log^{3/2}(m)} = O\left( \frac{\phi^{3/2}}{n^3 L^6 \log^{3/2}(m)} \right),$$

and obtain

$$\|\nabla_{\mathbf{W}_L} L(\widetilde{\mathbf{W}})\|_F \geq \frac{C_0 \phi^{1/2} m^{1/2} \sqrt{n \sum_{i=1}^{n} \|\mathbf{f}_{\widetilde{\mathbf{W}}}(\mathbf{x}_i) - \mathbf{y}_i\|_2^2}}{2\sqrt{k} n^2}.$$

Then plugging the fact that $1/(2n) \sum_{i=1}^{n} \|\mathbf{f}_{\widetilde{\mathbf{W}}}(\mathbf{x}_i) - \mathbf{y}_i\|_2^2 = L(\widetilde{\mathbf{W}})$, we are able to complete the proof. $\qquad\square$

## B.2 Proof of Lemma 4.2

*Proof of Lemma 4.2.* In order to prove Lemma 4.2, we first establish the function decrease of gradient descent. Note that we assume that all iterate are staying inside the region $\mathcal{B}(\mathbf{W}^{(0)}, \tau)$, then by Lemma 4.4, with probability at least $1 - \exp(-O(m\tau^{2/3}L))$, we have the following after one-step gradient descent

$$
\begin{aligned}
L(\mathbf{W}^{(t+1)}) \leq{}& L(\mathbf{W}^{(t)}) - \eta \|\nabla L(\mathbf{W}^{(t)})\|_F^2 \\
&+ C' \eta \sqrt{L(\mathbf{W}^{(t)})} \cdot \frac{\tau^{1/3} L^2 \sqrt{m \log(m)}}{\sqrt{k}} \cdot \|\nabla L(\mathbf{W}^{(t)})\|_2 + \frac{C'' L^2 m \eta^2}{k} \|\nabla L(\mathbf{W}^{(t)})\|_2^2.
\end{aligned}
$$
(B.1)

We first choose the step size

$$\eta = \frac{k}{4 C'' L^2 m} = O\left( \frac{k}{L^2 m} \right),$$

then (B.1) yields

$$
\begin{aligned}
L(\mathbf{W}^{(t+1)}) &\leq L(\mathbf{W}^{(t)}) - \frac{3\eta}{4} \|\nabla L(\mathbf{W}^{(t)})\|_F^2 + C' \eta \sqrt{L(\mathbf{W}^{(t)})} \cdot \frac{\tau^{1/3} L^2 \sqrt{m \log(m)}}{\sqrt{k}} \cdot \|\nabla L(\mathbf{W}^{(t)})\|_2 \\
&\leq L(\mathbf{W}^{(t)}) - \eta \|\nabla L(\mathbf{W}^{(t)})\|_F \left( \frac{3\|\nabla L(\mathbf{W}^{(t)})\|_F}{4} - C' \sqrt{L(\mathbf{W}^{(t)})} \cdot \frac{\tau^{1/3} L^2 \sqrt{m \log(m)}}{\sqrt{k}} \right),
\end{aligned}
$$

where we use the fact that $\|\nabla L(\mathbf{W}^{(t)})\|_2 \leq \|\nabla L(\mathbf{W}^{(t)})\|_F$. Then by Lemma 4.1, we know that with probability at least $1 - \exp\left( - O(m\phi/(kn)) \right)$

$$\|\nabla L(\mathbf{W}^{(t)})\|_F^2 \geq \|\nabla_{\mathbf{W}_L} L(\mathbf{W}^{(t)})\|_F^2 \geq \frac{Cm\phi}{kn^2} L(\mathbf{W}^{(t)}),$$
(B.2)

where $C$ is an absolute constant. Thus, we can choose the radius $\tau$ as

$$\tau = \frac{C^{3/2} \phi^{3/2}}{64 n^3 C'^3 L^6 \log^{3/2}(m)} = O\left( \frac{\phi^{3/2}}{n^3 L^6 \log^{3/2}(m)} \right),$$
(B.3)

and thus the following holds with probability at least $1 - \exp(-O(m\tau^{2/3}L)) - \exp\left( - O(m\phi/(kn)) \right) = 1 - \exp(-O(m\tau^{2/3}L))$,

$$L(\mathbf{W}^{(t+1)}) \leq L(\mathbf{W}^{(t)}) - \frac{\eta}{2} \|\nabla L(\mathbf{W}^{(t)})\|_F^2,$$
(B.4)

where the second inequality follows from (B.2). By triangle inequality, we have

$$\|\mathbf{W}_l^{(t)} - \mathbf{W}_l^{(0)}\|_2 \leq \sum_{s=0}^{t-1} \eta \|\nabla_{\mathbf{W}_l} L(\mathbf{W}^{(s)})\|_2 \leq \sum_{s=0}^{t-1} \eta \|\nabla L(\mathbf{W}^{(s)})\|_F.$$
(B.5)

Moreover, we have

$$\sqrt{L(\mathbf{W}^{(s)})} - \sqrt{L(\mathbf{W}^{(s+1)})} = \frac{L(\mathbf{W}^{(s)}) - L(\mathbf{W}^{(s+1)})}{\sqrt{L(\mathbf{W}^{(s)})} + \sqrt{L(\mathbf{W}^{(s+1)})}}$$

$$\geq \frac{\eta \|\nabla L(\mathbf{W}^{(s)})\|_F^2}{4\sqrt{L(\mathbf{W}^{(s)})}}$$

$$\geq \sqrt{\frac{Cm\phi}{kn^2}} \cdot \frac{\eta \|\nabla L(\mathbf{W}^{(s)})\|_F}{4},$$

where the second inequality is by (B.4) and the fact that $L(\mathbf{W}^{(s+1)}) \leq L(\mathbf{W}^{(s)})$, and the last inequality follows from (B.2). Plugging the above result into (B.5), we have with probability at least $1 - \exp(-O(m\tau^{2/3}L))$,

$$\|\mathbf{W}_l^{(t)} - \mathbf{W}_l^{(0)}\|_2 \leq \sum_{s=0}^{t-1} \eta \|\nabla L(\mathbf{W}^{(s)})\|_F$$

$$\leq 4\sqrt{\frac{kn^2}{Cm\phi}} \sum_{s=0}^{t-1} \left[ \sqrt{L(\mathbf{W}^{(s)})} - \sqrt{L(\mathbf{W}^{(s+1)})} \right]$$

$$\leq 4\sqrt{\frac{kn^2}{Cm\phi}} \cdot \sqrt{L(\mathbf{W}^{(0)})}. \tag{B.6}$$

Note that (B.6) holds for all $l$ and $t$. Then apply Lemma 4.3, we are able to complete the proof. $\qquad\square$

## B.3 Proof of Lemma 4.3

*Proof of Lemma 4.3.* Note that the output of the neural network can be formulated as

$$f_{\mathbf{W}^{(0)}}(\mathbf{x}_i) = \mathbf{V}\mathbf{x}_{L,i},$$

where $\mathbf{x}_{L,i}$ denotes the output of the last hidden layer with input $\mathbf{x}_i$. Note that each entry in $\mathbf{V}$ is i.i.d. generated from Gaussian distribution $\mathcal{N}(0, 1/k)$. Thus, we know that with probability at least $1 - \delta$, it holds that $\|\mathbf{V}\mathbf{x}_{L,i}\|_2 \leq \sqrt{\log(1/\delta)} \cdot \|\mathbf{x}_{L,i}\|_2$. Then by Lemma B.1 and union bound, we have $\|\mathbf{V}\mathbf{x}_{L,i}\|_2 \leq 2\sqrt{\log(1/\delta)}$ for all $i \in [n]$ with probability at least $1 - \exp(-O(m/L)) - n\delta$. Then we set $\delta = O(n^{-2})$ and use the fact that $m \geq O(L\log(nL))$, we have

$$f_{\mathbf{W}^{(0)}}(\mathbf{x}_i) = \|\mathbf{V}\mathbf{x}_{L,i}\|_2^2 \leq O(\log(n))$$

for all $i \in [n]$ with probability at least $1 - O(n^{-1})$. Then by our definition of training loss, it follows that

$$L(\mathbf{W}^{(0)}) = \frac{1}{2n} \sum_{i=1}^{n} \|\mathbf{f}_{\mathbf{W}^{(0)}}(\mathbf{x}_i) - \mathbf{y}_i\|_2^2$$

$$\leq \frac{1}{n} \sum_{i=1}^{n} \left[ \|\mathbf{f}_{\mathbf{W}^{(0)}}(\mathbf{x}_i)\|_2^2 + \|\mathbf{y}_i\|_2^2 \right]$$

$$\leq O(\log(n))$$

with probability at least $1 - O(n^{-1})$, where the first inequality is by Young's inequality and we assume that $\|\mathbf{y}_i\|_2 = O(1)$ for all $i \in [n]$ in the second inequality. This completes the proof. $\quad\square$

## B.4 Proof of Lemma 4.5

*Proof of Lemma 4.5.* By (B.4), we have

$$
\begin{aligned}
L\big(\mathbf{W}^{(t+1)}\big) &\leq L\big(\mathbf{W}^{(t)}\big) - \frac{\eta}{2}\|\nabla L(\mathbf{W}^{(t)})\|_F^2 \\
&\leq \left(1 - \frac{Cm\phi\eta}{2kn^2}\right) L\big(\mathbf{W}^{(t)}\big) \\
&\leq \left(1 - \frac{Cm\phi\eta}{2kn^2}\right)^{t+1} L\big(\mathbf{W}^{(0)}\big),
\end{aligned}
\tag{B.7}
$$

where the second inequality follows from (B.2). This completes the proof. $\qquad\square$

## B.5 Proof of Lemma A.2

*Proof of Lemma A.2.* Let $\mathbf{G}^{(t)}$ denote the stochastic gradient leveraged in the $t$-th iteration, where the corresponding minibatch is defined as $\mathcal{B}^{(t)}$. By Lemma 4.4, we have the following inequality regarding one-step stochastic gradient descent

$$
\begin{aligned}
L(\mathbf{W}^{(t+1)}) \leq{}& L(\mathbf{W}^{(t)}) - \eta\langle\nabla L(\mathbf{W}^{(t)}), \mathbf{G}^{(t)}\rangle \\
&+ C'\eta\sqrt{L\big(\mathbf{W}^{(t)}\big)} \cdot \frac{\tau^{1/3}L^2\sqrt{m\log(m)}}{\sqrt{k}} \cdot \|\mathbf{G}^{(t)}\|_2 + \frac{C''L^2m\eta^2}{k} \cdot \|\mathbf{G}^{(t)}\|_2^2.
\end{aligned}
$$

Then conditioned on $\mathbf{W}^{(t)}$, taking expectation over $\mathbf{G}^{(t)}$ on both sides gives

$$
\begin{aligned}
&\mathbb{E}\big[L(\mathbf{W}^{(t+1)})\big|\mathbf{W}^{(t)}\big] \\
&\leq L(\mathbf{W}^{(t)}) - \eta\|\nabla L(\mathbf{W}^{(t)})\|_F^2 + C'\eta\sqrt{L\big(\mathbf{W}^{(t)}\big)} \cdot \frac{\tau^{1/3}L^2\sqrt{m\log(m)}}{\sqrt{k}} \cdot \mathbb{E}\big[\|\mathbf{G}^{(t)}\|_2\big|\mathbf{W}^{(t)}\big] \\
&\quad + \frac{C''L^2m\eta^2}{k} \cdot \mathbb{E}\big[\|\mathbf{G}^{(t)}\|_2^2\big|\mathbf{W}^{(t)}\big].
\end{aligned}
\tag{B.8}
$$

Note that given $\mathbf{W}^{(t)}$, the expectations on $\|\mathbf{G}^{(t)}\|_2$ and $\|\mathbf{G}^{(t)}\|_2^2$ are only taken over the random minibatch $\mathcal{B}^{(t)}$. Similar to (A.1) and (A.2), we have

$$
\begin{aligned}
\mathbb{E}_{\mathcal{B}^{(t)}}[\|\mathbf{G}^{(t)} - \nabla L(\mathbf{W}^{(t)})\|_2^2] &\leq \frac{1}{B}\mathbb{E}_s[\|\nabla\ell\big(\mathbf{f}_{\mathbf{W}^{(t)}}(\mathbf{x}_s), \mathbf{y}_s\big) - \nabla L(\mathbf{W}^{(t)})\|_2^2] \\
&\leq \frac{2}{B}\big[\mathbb{E}_s[\|\nabla\ell\big(\mathbf{f}_{\mathbf{W}^{(t)}}(\mathbf{x}_s), \mathbf{y}_s\big)\|_2^2] + \|\nabla L(\mathbf{W}^{(t)})\|_2^2\big].
\end{aligned}
$$

Based on the definition of the $\ell_2$-norm of a collection of matrices and Lemma A.1, we have

$$
\mathbb{E}_s[\|\nabla\ell(f_{\mathbf{W}^{(t)}}(\mathbf{x}_s), y_s)\|_2^2] \leq \mathbb{E}_s\big[\max_{l\in[L]}\|\nabla_{\mathbf{W}_l}\ell(f_{\mathbf{W}^{(t)}}(\mathbf{x}_s), y_s)\|_2^2\big] \leq \frac{C_0 L(\mathbf{W}^{(t)})}{k},
$$

$$
\|L(\mathbf{W}^{(t)})\|_2^2 \leq \max_{l\in[L]}\|\nabla_{\mathbf{W}_l}L(\mathbf{W}^{(t)})\|_2^2 \leq \frac{C_0 L(\mathbf{W}^{(t)})}{k},
$$

where $C_0$ is an absolute constant. Then we have

$$
\begin{aligned}
\mathbb{E}[\|\mathbf{G}^{(t)}\|_2|\mathbf{W}^{(t)}]^2 &\leq \mathbb{E}[\|\mathbf{G}^{(t)}\|_2^2|\mathbf{W}^{(t)}] \\
&\leq 2\mathbb{E}_{\mathcal{B}^{(t)}}[\|\mathbf{G}^{(t)} - \nabla L(\mathbf{W}^{(t)})\|_2^2] + 2\|\nabla L(\mathbf{W}^{(t)})\|_2^2 \\
&\leq \frac{8C_0 m L(\mathbf{W}^{(t)})}{Bk} + 2\|\nabla L(\mathbf{W}^{(t)})\|_F^2.
\end{aligned}
$$

By (B.2), we know that there is a constant $C$ such that $\|\nabla L(\mathbf{W}^{(t)})\|_F^2 \geq Cm\phi L(\mathbf{W}^{(t)})/(kn^2)$. Then we set the step size $\eta$ and radius $\tau$ as follows

$$
\begin{aligned}
\eta &= \frac{Cd}{64C_0C''L^2mn^2} = O\left(\frac{1}{L^2mn^2}\right) \\
\tau &= \frac{C^3\phi^{3/2}B^3}{64^2n^6C_0^3C'^3L^6\log^{3/2}(m)} = O\left(\frac{\phi^{3/2}B^3}{n^6L^6\log^{3/2}(m)}\right)
\end{aligned}
\tag{B.9}
$$

Then (B.8) yields that

$$\mathbb{E}\big[L(\mathbf{W}^{(t+1)})\big|\mathbf{W}^{(t)}\big] \leq L\big(\mathbf{W}^{(t)}\big) - \eta\|\nabla L(\mathbf{W}^{(t)})\|_F^2 + \frac{C''L^2m\eta^2}{k}\left(\frac{8C_0n^2}{C\phi B}+2\right)\cdot\|\nabla L(\mathbf{W}^{(t)})\|_F^2$$

$$+ C'\eta\sqrt{L\big(\mathbf{W}^{(t)}\big)}\cdot\frac{\tau^{1/3}L^2\sqrt{m\log(m)}}{\sqrt{k}}\cdot\sqrt{\frac{8C_0n^2}{C\phi B}+2}\cdot\|\nabla L(\mathbf{W}^{(t)})\|_F$$

$$\leq L(\mathbf{W}^{(t)}) - \frac{\eta}{2}\|\nabla L(\mathbf{W}^{(t)})\|_F^2. \tag{B.10}$$

Then applying (B.2) again and taking expectation over $\mathbf{W}^{(t)}$ on both sides of (B.10), we obtain

$$\mathbb{E}\big[L(\mathbf{W}^{(t+1)})\big] \leq \left(1-\frac{Cm\phi\eta}{2kn^2}\right)\mathbb{E}[L(\mathbf{W}^{(t)})] \leq \left(1-\frac{Cm\phi\eta}{2kn^2}\right)^{t+1}L(\mathbf{W}^{(0)}).$$

This completes the proof.

$\square$

## B.6 Proof of Lemma A.3

*Proof of Lemma A.3.* We prove this by standard martingale inequality. By Lemma 4.4, we have

$$L(\mathbf{W}^{(t+1)}) \leq L(\mathbf{W}^{(t)}) - \eta\langle\nabla L(\mathbf{W}^{(t)}),\mathbf{G}^{(t)}\rangle$$

$$+ \eta C'\sqrt{L(\mathbf{W}^{(t)})}\cdot\frac{\tau^{1/3}L^2\sqrt{m\log(m)}}{\sqrt{k}}\cdot\|\mathbf{G}^{(t)}\|_2 + \frac{C''L^2m\eta^2}{k}\|\mathbf{G}^{(t)}\|_2^2.$$

Then by our choice of $\tau$ in (B.9), we have

$$\frac{\tau^{1/3}L^2\sqrt{m\log(m)}}{\sqrt{k}} = O\left(\frac{B\sqrt{m\phi}}{n^2\sqrt{k}}\right).$$

Note that by Lemma 4.1, we have

$$C'\sqrt{L(\mathbf{W}^{(t)})}\cdot\frac{\tau^{1/3}L^2\sqrt{m\log(m)}}{\sqrt{k}} = O\big(B\sqrt{m\phi L(\mathbf{W})/(kn^2)}/n\big) \leq \|\nabla L(\mathbf{W}^{(t)})\|_F.$$

Therefore, by inequality $\langle\mathbf{A},\mathbf{B}\rangle \leq \|\mathbf{A}\|_F\|\mathbf{B}\|_F$, we have

$$L(\mathbf{W}^{(t+1)}) \leq L(\mathbf{W}^{(t)}) + 2\eta\|\nabla L(\mathbf{W}^{(t)})\|_F\cdot\|\mathbf{G}^{(t)}\|_F + \frac{C''L^2m\eta^2}{k}\|\mathbf{G}^{(t)}\|_2^2. \tag{B.11}$$

By Lemma A.1, we know that there exists an absolute constant $C$ such that

$$\|\nabla L(\mathbf{W}^{(t)})\|_F^2 \leq \frac{CLmL(\mathbf{W}^{(t)})}{k} \text{ and } \|\mathbf{G}^{(t)}\|_F^2 \leq \frac{CLmnL(\mathbf{W}^{(t)})}{Bk},$$

where $B$ denotes the minibatch size and we use the fact that $\sum_{i\in\mathcal{B}^{(t)}}\ell(f_{\mathbf{W}^{(t)}}(\mathbf{x}_i),\mathbf{y}_i) \leq nL(\mathbf{W}^{(t)})$. Then note that $\eta \leq O\big(B^{1/2}k/(mL^2n^{1/2})\big)$, we have the following according to (B.11)

$$L(\mathbf{W}^{(t+1)}) \leq \left(1+\frac{C'Lmn^{1/2}\eta}{B^{1/2}k}\right)L(\mathbf{W}^{(t)}),$$

where $C'$ is an absolute constant. Taking logarithm on both sides further leads to

$$\log\big(L(\mathbf{W}^{(t+1)})\big) \leq \log\big(L(\mathbf{W}^{(t)})\big) + \frac{C'Lmn^{1/2}\eta}{B^{1/2}k},$$

where we use the inequality $\log(1+x) \leq x$. By (B.2) and (B.10), we know that

$$\mathbb{E}[L(\mathbf{W}^{(t+1)})|\mathbf{W}^{(t)}] \leq L(\mathbf{W}^{(t)}) - \frac{\eta}{2}\|\nabla L(\mathbf{W}^{(t)})\|_F^2 \leq \left(1-\frac{C''m\phi\eta}{kn^2}\right)L(\mathbf{W}^{(t)}).$$

Then by Jensen's inequality and the inequality $\log(1+x) \leq x$, we have

$$\mathbb{E}\big[\log\big(L(\mathbf{W}^{(t+1)})\big)|\mathbf{W}^{(t)}\big] \leq \log\big(\mathbb{E}[L(\mathbf{W}^{(t+1)})|\mathbf{W}^{(t)}]\big) \leq \log\big(L(\mathbf{W}^{(t)})\big) - \frac{C''m\phi\eta}{kn^2}.$$

Therefore we have $\{\log(L(\mathbf{W}^{(t)})) + C''m\phi t\eta/(kn^2)\}_{t=0,1...}$ is a super-martingale, and the martingale difference can be upper bounded by

$$\log(L(\mathbf{W}^{(t)})) + \frac{C''m\phi t\eta}{kn^2} - \log(L(\mathbf{W}^{(t-1)})) - \frac{C''m\phi(t-1)\eta}{kn^2} \leq \frac{C'Lmn^{1/2}\eta}{B^{1/2}k} + \frac{C''m\phi\eta}{kn^2}$$
$$\leq \frac{C'''Lmn^{1/2}\eta}{B^{1/2}k},$$

where $C'''$ is an absolute constant. By one-side Azuma's inequality for super-martingale, we know that for any $t$, with probability at least $1 - \delta$, the following holds

$$\log\big(L(\mathbf{W}^{(t)})\big) \leq \log\big(L(\mathbf{W}^{(0)})\big) - \frac{tC''m\phi\eta}{kn^2} + \frac{C'''Lmn^{1/2}\eta}{B^{1/2}k}\sqrt{2t\log(1/\delta)}$$
$$\leq \log\big(L(\mathbf{W}^{(0)})\big) - \frac{tC''m\phi\eta}{2kn^2} + \frac{C''''^2L^2mn^3\log(1/\delta)\eta}{C''kB\phi}, \qquad \text{(B.12)}$$

where the last inequality follows the fact that $-at + b\sqrt{t} \leq b^2/(4a)$ in the last inequality. Then we chose $\delta = O(m^{-1})$ and

$$\eta = \frac{\log(2)C''kB\phi}{C'^2L^2mn^3\log(1/\delta)} = O\bigg(\frac{kB\phi}{L^2n^3m\log(m)}\bigg).$$

Plugging these into (B.12) gives

$$\log\big(L(\mathbf{W}^{(t)})\big) \leq \log\big(2L(\mathbf{W}^{(0)})\big) - \frac{tC''m\phi\eta}{2kn^2},$$

which implies that

$$L(\mathbf{W}^{(t)}) \leq 2L(\mathbf{W}^{(0)}) \cdot \exp\bigg(-\frac{tC''m\phi\eta}{2kn^2}\bigg). \qquad \text{(B.13)}$$

By Lemma A.1 and the definition of $\mathbf{G}^{(t)}$, we have

$$\|\mathbf{G}^{(t)}\|_2 \leq O\bigg(\frac{m^{1/2}n^{1/2}\sqrt{L(\mathbf{W}^{(t)})}}{B^{1/2}k^{1/2}}\bigg) \qquad \text{(B.14)}$$

for all $t \leq T$. Therefore, plugging (B.14) into (B.13) and taking union bound over all $t \leq T$, and apply the result in Lemma 4.3, the following holds for all $t \leq T$ with probability at least $1 - O(T \cdot m^{-1}) - O(n^{-1}) = 1 - O(n^{-1})$,

$$\|\mathbf{W}_l^{(t)} - \mathbf{W}_l^{(0)}\|_2 \leq \sum_{s=0}^{t-1}\eta\|\mathbf{G}^{(t)}\|_2 \leq O\bigg(\frac{m^{1/2}n^{1/2}}{B^{1/2}k^{1/2}}\bigg) \cdot \sum_{s=0}^{t-1}\eta\sqrt{L(\mathbf{W}^{(s)})} \leq \widetilde{O}\bigg(\frac{k^{1/2}n^{5/2}}{B^{1/2}m^{1/2}\phi}\bigg),$$

where the first inequality is by triangle inequality, the second inequality follows from (B.14) and the last inequality is by (B.13) and Lemma 4.3. This completes the proof.

$\square$

## B.7   Proof of Lemma A.4

*Proof of Lemma A.4.* We first write the formula of $\nabla\ell\big(f_\mathbf{W}(\mathbf{x}_i), \mathbf{y}_i\big)$ as follows

$$\nabla\ell\big(f_\mathbf{W}(\mathbf{x}_i), \mathbf{y}_i\big) = \big[\big(f_\mathbf{W}(\mathbf{x}_i) - y_i\big)^\top\mathbf{V}\mathbf{\Sigma}_i\big]^\top\mathbf{x}_i^\top.$$

Since $\mathbf{\Sigma}_i$ is an diagonal matrix with $\big(\mathbf{\Sigma}_i\big)_{jj} = \sigma'(\langle\mathbf{w}_j, \mathbf{x}_i\rangle)$. Therefore, it holds that

$$\|\nabla\ell(f_\mathbf{W}(\mathbf{x}_i), \mathbf{y}_i)\|_{2,\infty} = \max_{j\in[m]}\langle f_\mathbf{W}(\mathbf{x}_i) - \mathbf{y}_i, \mathbf{v}_j\rangle \cdot \|\mathbf{x}_i\|_2 \leq \max_{j\in[m]}\|f_\mathbf{W}(\mathbf{x}_i) - \mathbf{y}_i\|_2\|\mathbf{v}_j\|_2,$$

(B.15)

where $\mathbf{v}_j \in \mathbb{R}^k$ denotes the $j$-th column of $\mathbf{V}$ and we use the fact that $\|\mathbf{x}_i\|_2 = 1$. Note that $\mathbf{v}_j \sim \mathcal{N}(0, \mathbf{I}/k)$, we have

$$\mathbb{P}\big(\|\mathbf{v}_j\|_2^2 \geq O\big(\log(m)\big)\big) \leq O(m^{-c}),$$

for any positive constant $c$. Setting $c = 2$ and applying union bound over $\mathbf{v}_1, \ldots, \mathbf{v}_m$, we have with probability at least $1 - O(m^{-1})$,

$$\max_{j \in [m]} \|\mathbf{v}_j\|_2 \leq O\big(\log^{1/2}(m)\big).$$

Plugging this into (B.15) and applying the fact that $\|f_{\mathbf{W}}(\mathbf{x}_i) - \mathbf{y}_i\|_2 = \sqrt{\ell(f_{\mathbf{W}}(\mathbf{x}_i), \mathbf{y}_i)}$, we are able to complete the proof. $\qquad\square$

## B.8  Proof of Lemma A.5

Recall that the output of two-layer ReLU network can be formulated as

$$\mathbf{f}_{\mathbf{W}}(\mathbf{x}_i) = \mathbf{V}\boldsymbol{\Sigma}_i\mathbf{W}\mathbf{x}_i,$$

where $\boldsymbol{\Sigma}_i$ is a diagonal matrix with only non-zero diagonal entry $(\boldsymbol{\Sigma}_i)_{jj} = \sigma'(\mathbf{w}_j^\top\mathbf{x}_i)$. Then based on the definition of $L(\mathbf{W})$, we have

$$
\begin{aligned}
&L(\widetilde{\mathbf{W}}) - L(\widehat{\mathbf{W}}) \\
&= \frac{1}{2n} \sum_{i=1}^n \|\mathbf{V}\widetilde{\boldsymbol{\Sigma}}_i\widetilde{\mathbf{W}}\mathbf{x}_i - \mathbf{y}_i\|_2^2 - \frac{1}{2n} \sum_{i=1}^n \|\mathbf{V}\widehat{\boldsymbol{\Sigma}}_i\widehat{\mathbf{W}}\mathbf{x}_i - \mathbf{y}_i\|_2^2 \\
&= \underbrace{\frac{1}{2n} \sum_{i=1}^n \big\langle \mathbf{V}\widehat{\boldsymbol{\Sigma}}_i\widehat{\mathbf{W}}\mathbf{x}_i - \mathbf{y}_i, \mathbf{V}\widetilde{\boldsymbol{\Sigma}}_i\widetilde{\mathbf{W}}\mathbf{x}_i - \mathbf{V}\widehat{\boldsymbol{\Sigma}}_i\widehat{\mathbf{W}}\mathbf{x}_i\big\rangle}_{I_1} + \underbrace{\frac{1}{2n} \sum_{i=1}^n \big\|\mathbf{V}\widetilde{\boldsymbol{\Sigma}}_i\widetilde{\mathbf{W}}\mathbf{x}_i - \mathbf{V}\widehat{\boldsymbol{\Sigma}}_i\widehat{\mathbf{W}}\mathbf{x}_i\big\|_2^2}_{I_2}.
\end{aligned}
$$

Then we tackle the two terms on the R.H.S. of the above equation separately. Regarding the first term, i.e., $I_1$, we have

$$
\begin{aligned}
I_1 &= \frac{1}{2n} \sum_{i=1}^n \big\langle \mathbf{V}\widehat{\boldsymbol{\Sigma}}_i\widehat{\mathbf{W}}\mathbf{x}_i - \mathbf{y}_i, \mathbf{V}\widehat{\boldsymbol{\Sigma}}_i(\widetilde{\mathbf{W}} - \widehat{\mathbf{W}})\mathbf{x}_i\big\rangle \\
&\quad + \frac{1}{2n} \sum_{i=1}^n \big\langle \mathbf{V}\widehat{\boldsymbol{\Sigma}}_i\widehat{\mathbf{W}}\mathbf{x}_i - \mathbf{y}_i, \mathbf{V}(\widetilde{\boldsymbol{\Sigma}}_i - \widehat{\boldsymbol{\Sigma}}_i)\widetilde{\mathbf{W}}\mathbf{x}_i\big\rangle \\
&\leq \big\langle \nabla L(\widehat{\mathbf{W}}), \widetilde{\mathbf{W}} - \widehat{\mathbf{W}}\big\rangle + \frac{1}{2n} \sum_{i=1}^n \sqrt{\ell(f_{\widehat{\mathbf{W}}}(\mathbf{x}_i), \mathbf{y}_i)} \cdot \|\mathbf{V}(\widetilde{\boldsymbol{\Sigma}}_i - \widehat{\boldsymbol{\Sigma}}_i)\widetilde{\mathbf{W}}\mathbf{x}_i\|_2.
\end{aligned}
$$

Note that the non-zero entries in $\widetilde{\boldsymbol{\Sigma}}_i - \widehat{\boldsymbol{\Sigma}}_i$ represent the nodes, say $j$, satisfying $\text{sign}(\widetilde{\mathbf{w}}_j^\top\mathbf{x}_i) \neq \text{sign}(\widehat{\mathbf{w}}_j^\top\mathbf{x}_i)$, which implies $\big|\widetilde{\mathbf{w}}_j^\top\mathbf{x}_i\big| \leq \big|(\widetilde{\mathbf{w}}_j - \widehat{\mathbf{w}}_j)^\top\mathbf{x}_i\big|$. Therefore, we have

$$\|\mathbf{V}(\widetilde{\boldsymbol{\Sigma}}_i - \widehat{\boldsymbol{\Sigma}}_i)\widetilde{\mathbf{W}}\mathbf{x}_i\|_2^2 \leq \|\mathbf{V}(\widetilde{\boldsymbol{\Sigma}}_i - \widehat{\boldsymbol{\Sigma}}_i)(\widetilde{\mathbf{W}} - \widehat{\mathbf{W}})\mathbf{x}_i\|_2^2.$$

By Lemma B.3, we have $\|\widetilde{\boldsymbol{\Sigma}}_i - \widehat{\boldsymbol{\Sigma}}_i\|_0 \leq \|\widetilde{\boldsymbol{\Sigma}}_i - \boldsymbol{\Sigma}_i\|_0 + \|\widehat{\boldsymbol{\Sigma}}_i - \boldsymbol{\Sigma}_i\|_0 = O(m\tau^{2/3})$. Then we define $\bar{\boldsymbol{\Sigma}}_i$ as

$$\big(\bar{\boldsymbol{\Sigma}}_i\big)_{jk} = |\big(\widetilde{\boldsymbol{\Sigma}}_i - \widehat{\boldsymbol{\Sigma}}_i\big)_{jk}| \quad \text{for all } j, k.$$

Then we have

$$
\begin{aligned}
\|\mathbf{V}(\widetilde{\boldsymbol{\Sigma}}_i - \widehat{\boldsymbol{\Sigma}}_i)\widetilde{\mathbf{W}}\mathbf{x}_i\|_2 &\leq \|\mathbf{V}(\widetilde{\boldsymbol{\Sigma}}_i - \widehat{\boldsymbol{\Sigma}}_i)\bar{\boldsymbol{\Sigma}}_i(\widetilde{\mathbf{W}} - \widehat{\mathbf{W}})\mathbf{x}_i\|_2 \\
&\leq \|\mathbf{V}(\widetilde{\boldsymbol{\Sigma}}_i - \widehat{\boldsymbol{\Sigma}}_i)\|_2 \cdot \|\bar{\boldsymbol{\Sigma}}_i(\widetilde{\mathbf{W}} - \widehat{\mathbf{W}})\|_F \\
&= \|\mathbf{V}(\widetilde{\boldsymbol{\Sigma}}_i - \widehat{\boldsymbol{\Sigma}}_i)\|_2 \cdot \sqrt{\sum_{j=1}^m (\bar{\boldsymbol{\Sigma}}_i)_{jj}\|\widetilde{\mathbf{w}}_j - \widehat{\mathbf{w}}_j\|_2^2} \\
&\leq \|\mathbf{V}(\widetilde{\boldsymbol{\Sigma}}_i - \widehat{\boldsymbol{\Sigma}}_i)\|_2 \cdot \|\bar{\boldsymbol{\Sigma}}_i\|_0^{1/2} \cdot \|\widetilde{\mathbf{W}} - \widehat{\mathbf{W}}\|_{2,\infty},
\end{aligned}
$$

where $\widetilde{\mathbf{w}}_j$ and $\widehat{\mathbf{w}}_j$ denote the $j$-th columns of $\widetilde{\mathbf{W}}$ and $\widehat{\mathbf{W}}$ respectively. By Lemma B.3 and the fact that $\|\bar{\mathbf{\Sigma}}_i\|_0 = O(m\tau^{2/3})$, we have with probability $1 - O(m\tau^{2/3})$

$$\|\mathbf{V}(\widetilde{\mathbf{\Sigma}}_i - \widehat{\mathbf{\Sigma}}_i)\widetilde{\mathbf{W}}\mathbf{x}_i\|_2 \le O(m\sqrt{\log(m)}\tau^{2/3}k^{-1}) \cdot \|\widetilde{\mathbf{W}} - \widehat{\mathbf{W}}\|_{2,\infty}. \tag{B.16}$$

Therefore, we have

$$I_1 \le \langle \nabla L(\widehat{\mathbf{W}}), \widetilde{\mathbf{W}} - \widehat{\mathbf{W}}\rangle + \frac{1}{2n}\sum_{i=1}^{n}\sqrt{\ell(f_{\widehat{\mathbf{W}}}(\mathbf{x}_i), \mathbf{y}_i)} \cdot \|\mathbf{V}(\widetilde{\mathbf{\Sigma}}_i - \widehat{\mathbf{\Sigma}}_i)\widetilde{\mathbf{W}}\mathbf{x}_i\|_2$$

$$\le \langle \nabla L(\widehat{\mathbf{W}}), \widetilde{\mathbf{W}} - \widehat{\mathbf{W}}\rangle + O(m\sqrt{\log(m)}\tau^{2/3}k^{-1/2}) \cdot \sqrt{L(\widehat{\mathbf{W}})} \cdot \|\widetilde{\mathbf{W}} - \widehat{\mathbf{W}}\|_{2,\infty},$$

where the last inequality follows from (B.16) and Young's inequality. In what follows we are going to tackle the term $I_2$. Note that for each $i$, we have

$$\|\mathbf{V}\widetilde{\mathbf{\Sigma}}_i\widetilde{\mathbf{W}}\mathbf{x}_i - \mathbf{V}\widehat{\mathbf{\Sigma}}_i\widehat{\mathbf{W}}\mathbf{x}_i\|_2 = \|\mathbf{V}\widehat{\mathbf{\Sigma}}_i(\widetilde{\mathbf{W}} - \widehat{\mathbf{W}})\mathbf{x}_i\|_2 + \|\mathbf{V}(\widetilde{\mathbf{\Sigma}}_i - \widehat{\mathbf{\Sigma}}_i)\widetilde{\mathbf{W}}\mathbf{x}_i\|_2$$

$$\le \|\mathbf{V}\|_2\|\widetilde{\mathbf{W}} - \widehat{\mathbf{W}}\|_2 + \|\mathbf{V}(\widetilde{\mathbf{\Sigma}}_i - \widehat{\mathbf{\Sigma}}_i)\|_2 \cdot \|\widetilde{\mathbf{W}} - \widehat{\mathbf{W}}\|_2$$

$$= O(m^{1/2}/k^{1/2}) \cdot \|\widetilde{\mathbf{W}} - \widehat{\mathbf{W}}\|_2,$$

where the last inequality holds due to the fact that $\|\mathbf{V}\|_2 = O(m^{1/2}/k^{1/2})$ with probability at least $1 - \exp(-O(m/k))$. This leads to $I_2 \le O(m/k) \cdot \|\widetilde{\mathbf{W}} - \widehat{\mathbf{W}}\|_2^2$. Now we can put everything together, and obtain

$$L(\widetilde{\mathbf{W}}) - L(\widehat{\mathbf{W}}) = I_1 + I_2$$

$$\le \langle \nabla L(\widehat{\mathbf{W}}), \widetilde{\mathbf{W}} - \widehat{\mathbf{W}}\rangle + O(m\sqrt{\log(m)}\tau^{2/3}k^{-1/2}) \cdot \sqrt{L(\widehat{\mathbf{W}})} \cdot \|\widetilde{\mathbf{W}} - \widehat{\mathbf{W}}\|_{2,\infty}$$

$$+ O(m/k) \cdot \|\widetilde{\mathbf{W}} - \widehat{\mathbf{W}}\|_2^2.$$

Then applying union bound on the inequality for $I_1$ and $I_2$, we are able to complete the proof.

## C  Proof of Technical Lemmas in Appendix B

### C.1  Proof of Lemma B.2

Let $\mathbf{z}_1, \ldots, \mathbf{z}_n \in \mathbb{R}^d$ be $n$ vectors with $1/2 \le \min_i\{\|\mathbf{z}_i\|_2\} \le \max_i\{\|\mathbf{z}_i\|_2\} \le 2$. Let $\bar{\mathbf{z}}_i = \mathbf{z}_i/\|\mathbf{z}_i\|_2$ and assume $\min_{i,j}\|\bar{\mathbf{z}}_i - \bar{\mathbf{z}}_j\|_2, \min_{i,j}\|\bar{\mathbf{z}}_i + \bar{\mathbf{z}}_j\|_2 \ge \widetilde{\phi}$. For each $\mathbf{z}_i$, we construct an orthonormal matrix $\mathbf{Q}_i = [\bar{\mathbf{z}}_i, \mathbf{Q}_i'] \in \mathbb{R}^{d \times d}$. Then consider a random vector $\mathbf{w} \in \mathbb{R}^d$ following distribution $\mathcal{N}(0, \mathbf{I})$, it follows that $\mathbf{u}_i := \mathbf{Q}_i^\top \mathbf{w} \sim \mathcal{N}(0, \mathbf{I})$. Then we can decompose $\mathbf{w}$ as

$$\mathbf{w} = \mathbf{Q}_i\mathbf{u}_i = \mathbf{u}_i^{(1)}\bar{\mathbf{z}}_i + \mathbf{Q}_i'\mathbf{u}_i', \tag{C.1}$$

where $\mathbf{u}_i^{(1)}$ denotes the first coordinate of $\mathbf{u}_i$ and $\mathbf{u}_i' := [\mathbf{u}_i^{(2)}, \ldots, \mathbf{u}_i^{(d)}]^\top$. Then let $\gamma = \sqrt{\pi}\widetilde{\phi}/(8n)$, we define the following set of $\mathbf{w}$ based on $\mathbf{z}_i$,

$$\mathcal{W}_i = \left\{\mathbf{w} : |\mathbf{u}_i^{(1)}| \le \gamma, |\langle \mathbf{Q}_i'\mathbf{u}_i', \bar{\mathbf{z}}_j\rangle| \ge 2\gamma \text{ for all } \bar{\mathbf{z}}_j \text{ such that } j \neq i\right\}.$$

Regarding the class of sets $\{\mathcal{W}_1, \ldots, \mathcal{W}_n\}$, we have the following lemmas.

**Lemma C.1.** For any $\mathcal{W}_i$ and $\mathcal{W}_j$ with $i \neq j$, we have

$$\mathbb{P}(\mathbf{w} \in \mathcal{W}_i) \ge \frac{\widetilde{\phi}}{n\sqrt{128e}} \quad \text{and} \quad \mathcal{W}_i \cap \mathcal{W}_j = \emptyset.$$

Then we deliver the following two lemmas which are useful to establish the required lower bound.

**Lemma C.2.** For any $\mathbf{a} = (a_1, \ldots, a_n)^\top \in \mathbb{R}^n$, let $\mathbf{h}(\mathbf{w}) = \sum_{i=1}^{n} a_i\sigma'(\langle \mathbf{w}, \mathbf{z}_i\rangle)\mathbf{z}_i$ where $\mathbf{w} \sim N(\mathbf{0}, \mathbf{I})$ is a Gaussian random vector. Then it holds that

$$\mathbb{P}\big[\|\mathbf{h}(\mathbf{w})\|_2 \ge |a_i|/4 \big| \mathbf{w} \in \mathcal{W}_i\big] \ge 1/2.$$

Now we are able to prove Lemma B.2.

*Proof of Lemma B.2.* We first prove the result for any fixed $\mathbf{u}_1, \ldots, \mathbf{u}_n$. Then we define $a_i(\mathbf{v}_j) = \langle \mathbf{u}_i, \mathbf{v}_j \rangle$, $\mathbf{w}_j = \sqrt{m/2}\mathbf{w}_{L,j}^{(0)}$ and

$$\mathbf{h}(\mathbf{v}_j, \mathbf{w}_j) = \sum_{i=1}^n a_i(\mathbf{v}_j)\sigma'(\langle \mathbf{w}_j, \mathbf{x}_{L-1,i} \rangle)\mathbf{x}_{L-1,i}.$$

Then we define the event

$$\mathcal{E}_i = \left\{ j \in [m] : \mathbf{w}_j' \in \mathcal{W}_i, \|\mathbf{h}(\mathbf{v}_j, \mathbf{w}_j)\|_2 \geq |a_i(\mathbf{v}_j)|/4, |a_i(\mathbf{v}_j)| \geq \|\mathbf{u}_i\|_2/\sqrt{k} \right\}.$$

By Lemma B.1, we know that with high probability $1/2 \leq \|\mathbf{x}_{L-1,i}\|_2 \leq 2$ for all $i$ and $\big\|\mathbf{x}_{L-1,i}/\|\mathbf{x}_{L-1,i}\|_2 - \mathbf{x}_{L-1,j}/\|\mathbf{x}_{L-1,j}\|_2\big\| \geq \phi/2$ and $\big\|\mathbf{x}_{L-1,i}/\|\mathbf{x}_{L-1,i}\|_2 + \mathbf{x}_{L-1,j}/\|\mathbf{x}_{L-1,j}\|_2\big\| \geq \phi/2$ for all $i \neq j$. Then by Lemmas C.1 and C.2 we know that $\mathcal{E}_i \cap \mathcal{E}_j = \emptyset$ if $i \neq j$ and

$$\mathbb{P}(j \in \mathcal{E}_i) = \mathbb{P}\big[\|\mathbf{h}(\mathbf{v}_j, \mathbf{w}_j)\|_2 \geq |a_i(\mathbf{v}_j)|/4 | \mathbf{w}_j' \in \mathcal{W}_i\big] \cdot \mathbb{P}\big[\mathbf{w}_j' \in \mathcal{W}_i\big] \cdot \mathbb{P}\big[|a_i(\mathbf{v}_j)| \geq \|\mathbf{u}_i\|_2/\sqrt{k}\big]$$

$$\geq \frac{\phi}{64\sqrt{2}en}, \tag{C.2}$$

where the first equality holds because $\mathbf{w}_j$ and $\mathbf{v}_j$ are independent, and the second inequality follows from Lemmas C.1, C.2 and the fact that $\mathbb{P}(|a_i(\mathbf{v}_j)| \geq \|\mathbf{u}_i\|_2/\sqrt{k}) \geq 1/2$. Then we have

$$\|\nabla_{\mathbf{W}_L}L(\mathbf{W})\|_F^2 = \frac{1}{n^2}\sum_{j=1}^m \|\mathbf{h}(\mathbf{v}_j, \mathbf{w}_j)\|_2^2$$

$$\geq \frac{1}{n^2}\sum_{j=1}^m \|\mathbf{h}(\mathbf{v}_j, \mathbf{w}_j)\|_2^2\sum_{s=1}^n \mathbb{1}\big(j \in \mathcal{E}_s\big)$$

$$\geq \frac{1}{n^2}\sum_{j=1}^m\sum_{s=1}^n \frac{\|\mathbf{u}_s\|_2^2}{16k}\mathbb{1}\big(j \in \mathcal{E}_s\big),$$

where the second inequality holds due to the fact that

$$\|\mathbf{h}(\mathbf{v}_j, \mathbf{w}_j)\|_2^2\mathbb{1}\big(j \in \mathcal{E}_s\big) \geq \frac{a_s^2(\mathbf{v}_j)}{16}\mathbb{1}(|a_s(\mathbf{v}_j)| \geq \|\mathbf{u}_s\|_2/\sqrt{k}) \cdot \mathbb{1}(j \in \mathcal{E}_s)$$

$$\geq \frac{\|\mathbf{u}_s\|_2^2}{16k}\mathbb{1}(j \in \mathcal{E}_s),$$

where the first inequality follows from the definition of $\mathcal{E}_s$. Then we further define

$$Z_j = \sum_{s=1}^n \frac{\|\mathbf{u}_s\|_2^2}{16k}\mathbb{1}\big(j \in \mathcal{E}_s\big),$$

and provide the following results for $\mathbb{E}[Z(\mathbf{w}_j)]$ and $\text{var}[Z(\mathbf{w}_j)]$

$$\mathbb{E}[Z_j] = \sum_{s=1}^n \frac{\|\mathbf{u}_s\|_2^2}{16k}\mathbb{P}\big(j \in \mathcal{E}_s\big), \qquad \text{var}[Z(\mathbf{w})] = \sum_{s=1}^n \frac{\|\mathbf{u}_s\|_2^4}{256k^2}\mathbb{P}\big(j \in \mathcal{E}_s\big)\big[1 - \mathbb{P}\big(j \in \mathcal{E}_s\big)\big].$$

Then by Bernstein inequality, with probability at least $1 - \exp\big(-O\big(m\mathbb{E}[Z(\mathbf{w})]/\max_{i \in [n]}\|\mathbf{u}_i\|_2^2\big)\big)$, it holds that

$$\sum_{j=1}^m Z_j \geq \frac{m}{2}\mathbb{E}[Z_j] \geq \sum_{i=1}^n \frac{\|\mathbf{u}_i\|_2^2}{32k} \cdot \frac{m\phi}{64\sqrt{2}en} = \frac{C\phi m\sum_{i=1}^n \|\mathbf{u}_i\|_2^2}{kn},$$

where the second inequality follows from (C.2) and $C = 1/(2096\sqrt{2}e)$ is an absolute constant. Therefore, with probability at least $1 - \exp\big(-O(m\phi/(kn))\big)$ we have

$$\sum_{j=1}^m \left\|\frac{1}{n}\sum_{i=1}^n \langle \mathbf{u}_i, \mathbf{v}_j \rangle \sigma'\big(\langle \mathbf{w}_{L,j}, \mathbf{x}_{L-1,i} \rangle\big)\mathbf{x}_{L-1,i}\right\|_2^2 \geq \frac{1}{n^2}\sum_{j=1}^m Z(\mathbf{w}_j) \geq \frac{C\phi m\sum_{i=1}^n \|\mathbf{u}_i\|_2^2}{kn^3}.$$

Till now we have completed the proof for one particular vector collection $\{\mathbf{u}_i\}_{i=1,\ldots,n}$. Then we are going to prove that the above inequality holds for arbitrary $\{\mathbf{u}_i\}_{i=1,\ldots,n}$ with high probability. Taking $\epsilon$-net over all possible vectors $\{\mathbf{u}_1, \ldots, \mathbf{u}_n\} \in (\mathbb{R}^k)^n$ and applying union bound, the above inequality holds with probability at least $1 - \exp\big(-O(m\phi/(kn)) + \widetilde{O}(nk)\big)$. Since we have $m \geq \widetilde{O}\big(\phi^{-1}n^2k^2\big)$, the desired result holds for all choices of $\{\mathbf{u}_1, \ldots, \mathbf{u}_n\}$.

$\square$

# D    Proof of Auxiliary Lemmas in Appendix C

*Proof of Lemma C.1.* We first prove that any two sets $\mathcal{W}_i$ and $\mathcal{W}_j$ have not overlap region. Consider an vector $\mathbf{w} \in \mathcal{W}_i$ with the decomposition

$$\mathbf{w} = \mathbf{u}_i^{(1)}\bar{\mathbf{z}}_i + \mathbf{Q}_i'\mathbf{u}_i'.$$

Then based on the definition of $\mathcal{W}_i$ we have,

$$\langle \mathbf{w}, \bar{\mathbf{z}}_j \rangle = \langle \mathbf{u}_i^{(1)}\bar{\mathbf{z}}_i + \mathbf{Q}_i'\mathbf{u}_i', \bar{\mathbf{z}}_j \rangle = \mathbf{u}_i^{(1)}\langle \bar{\mathbf{z}}_i, \bar{\mathbf{z}}_j \rangle + \langle \mathbf{Q}_i'\mathbf{u}_i', \bar{\mathbf{z}}_j \rangle.$$

Since $\mathbf{w} \in \mathcal{W}_i$, we have $|\mathbf{u}_i^{(1)}| \leq \gamma$ and $|\langle \mathbf{Q}'\mathbf{u}_i', \bar{\mathbf{z}}_j \rangle| \geq 2\gamma$. Therefore, note that $|\langle \bar{\mathbf{z}}_i, \bar{\mathbf{z}}_j \rangle| \leq 1$, it holds that

$$|\langle \mathbf{w}, \bar{\mathbf{z}}_j \rangle| \geq \big||\langle \mathbf{Q}_i'\mathbf{u}_i', \bar{\mathbf{z}}_j \rangle| - |\mathbf{u}_i^{(1)}|\big| > \gamma. \tag{D.1}$$

Note that set $\mathcal{W}_j$ requires $|\mathbf{u}_j^{(1)}| = \langle \mathbf{w}, \bar{\mathbf{z}}_j \rangle \leq \gamma$, which conflicts with (D.1). This immediately implies that $\mathcal{W}_i \cap \mathcal{W}_j = \emptyset$.

Then we are going to compute the probability $\mathbb{P}(\mathbf{w} \in \mathcal{W}_i)$. Based on the parameter $\gamma$, we define the following two events

$$\mathcal{E}_1(\gamma) = \big\{|\mathbf{u}_i^{(1)}| \leq \gamma\big\}, \ \mathcal{E}_2(\gamma) = \big\{|\langle \mathbf{Q}_i'\mathbf{u}_i', \bar{\mathbf{z}}_j \rangle| \geq 2\gamma \text{ for all } \bar{\mathbf{z}}_j, j \neq i\big\}.$$

Evidently, we have $\mathbb{P}(\mathbf{w} \in \mathcal{W}_i) = \mathbb{P}(\mathcal{E}_1)\mathbb{P}(\mathcal{E}_2)$. Since $\mathbf{u}_i^{(1)}$ is a standard Gaussian random variable, we have

$$\mathbb{P}(\mathcal{E}_1) = \frac{1}{\sqrt{2\pi}} \int_{-\gamma}^{\gamma} \exp\Big(-\frac{1}{2}x^2\Big) \mathrm{d}x \geq \sqrt{\frac{2}{\pi e}}\gamma.$$

Moreover, by definition, for any $j = 1, \dots, n$ we have

$$\langle \mathbf{Q}_i'\mathbf{u}_i', \bar{\mathbf{z}}_j \rangle \sim N\big[0, 1 - (\bar{\mathbf{z}}_i^{\top}\bar{\mathbf{z}}_j)^2\big].$$

Note that for any $j \neq i$ we have $\|\bar{\mathbf{z}}_i - \bar{\mathbf{z}}_j\|_2 \geq \widetilde{\phi}$ and $\|\bar{\mathbf{z}}_i + \bar{\mathbf{z}}_j\|_2 \geq \widetilde{\phi}$, then it follows that

$$|\langle \bar{\mathbf{z}}_i, \bar{\mathbf{z}}_j \rangle| \leq 1 - \widetilde{\phi}^2/2,$$

and if $\widetilde{\phi}^2 \leq 2$, then

$$1 - (\bar{\mathbf{z}}_i^{\top}\bar{\mathbf{z}}_j)^2 \geq \widetilde{\phi}^2 - \widetilde{\phi}^4/4 \geq \widetilde{\phi}^2/2.$$

Therefore for any $j \neq i$,

$$\mathbb{P}[|\langle \mathbf{Q}_i'\mathbf{u}_i', \bar{\mathbf{z}}_j \rangle| < 2\gamma] = \frac{1}{\sqrt{2\pi}} \int_{-2[1-(\bar{\mathbf{z}}_i^{\top}\bar{\mathbf{z}}_j)^2]^{-1/2}\gamma}^{2[1-(\bar{\mathbf{z}}_i^{\top}\bar{\mathbf{z}}_j)^2]^{-1/2}\gamma} \exp\Big(-\frac{1}{2}x^2\Big) \mathrm{d}x \leq \sqrt{\frac{8}{\pi}}\frac{\gamma}{[1-(\bar{\mathbf{z}}_i^{\top}\bar{\mathbf{z}}_j)^2]^{1/2}} \leq \frac{4}{\sqrt{\pi}}\gamma\widetilde{\phi}^{-1}.$$

By union bound over $[n]$, we have

$$\mathbb{P}(\mathcal{E}_2) = \mathbb{P}[|\langle \mathbf{Q}_i'\mathbf{u}_i', \bar{\mathbf{z}}_j \rangle| \geq 2\gamma, j \in \mathcal{I}] \geq 1 - \frac{4}{\sqrt{\pi}}n\gamma\widetilde{\phi}^{-1}.$$

Therefore we have

$$\mathbb{P}(\mathbf{w} \in \mathcal{W}_i) \geq \sqrt{\frac{2}{\pi e}}\gamma \cdot \Big(1 - \frac{4}{\sqrt{\pi}}n\gamma\widetilde{\phi}^{-1}\Big).$$

Plugging $\gamma = \sqrt{\pi}\widetilde{\phi}/(8n)$, it holds that $\mathbb{P}(\mathcal{E}) \geq \widetilde{\phi}/(\sqrt{128e}n)$. This completes the proof. $\quad\square$

*Proof of Lemma C.2.* Recall the decomposition of $\mathbf{w}$ in (C.1),

$$\mathbf{w} = \mathbf{u}_i^{(1)}\bar{\mathbf{z}}_i + \mathbf{Q}_i'\mathbf{u}_i'.$$

Define the event $\mathcal{E}_i := \{\mathbf{w} \in \mathcal{W}_i\}$. Then conditioning on $\mathcal{E}_i$, we have

$$\begin{aligned}
\mathbf{h}(\mathbf{w}) &= \sum_{i=1}^{n} a_i \sigma'(\langle \mathbf{w}, \mathbf{z}_i \rangle) \mathbf{z}_i \\
&= a_i \sigma'(\mathbf{u}_i^{(1)}) \mathbf{z}_i + \sum_{j \neq i} a_j \sigma'\big(\mathbf{u}_i^{(1)} \langle \bar{\mathbf{z}}_i, \bar{\mathbf{z}}_j \rangle + \langle \mathbf{Q}_i' \mathbf{u}_i', \bar{\mathbf{z}}_j \rangle\big) \mathbf{z}_j \\
&= a_i \sigma'(\mathbf{u}_i^{(1)}) \mathbf{z}_i + \sum_{j \neq i} a_j \sigma'\big(\langle \mathbf{Q}_i' \mathbf{u}_i', \mathbf{z}_j \rangle\big) \mathbf{z}_j
\end{aligned} \tag{D.2}$$

where the last equality follows from the fact that conditioning on event $\mathcal{E}_i$, for all $j \neq i$, it holds that $|\langle \mathbf{Q}_i' \mathbf{u}_i', \bar{\mathbf{z}}_j \rangle| \geq 2\gamma > |\mathbf{u}_i^{(1)}| \geq |\mathbf{u}_i^{(1)} \langle \bar{\mathbf{z}}_i, \bar{\mathbf{z}}_j \rangle|$. We then consider two cases: $\mathbf{u}_i^{(1)} > 0$ and $\mathbf{u}_i^{(1)} < 0$, which occur equally likely conditioning on the event $\mathcal{E}_i$. Let $u_1 > 0$ and $u_2 < 0$ denote $\mathbf{u}_i^{(1)}$ in these two cases, we have

$$\mathbb{P}\bigg[ \|\mathbf{h}(\mathbf{w})\|_2 \geq \inf_{u_1 > 0, u_2 < 0} \max\big\{ \|\mathbf{h}(u_1 \bar{\mathbf{z}}_i + \mathbf{Q}_i' \mathbf{u}_i')\|_2, \|\mathbf{h}(u_2 \bar{\mathbf{z}}_i + \mathbf{Q}_i' \mathbf{u}_i')\|_2 \big\} \bigg| \mathcal{E}_i \bigg] \geq 1/2.$$

By the inequality $\max\{\|\mathbf{a}\|_2, \|\mathbf{b}\|_2\} \geq \|\mathbf{a} - \mathbf{b}\|_2/2$, we have

$$\mathbb{P}\bigg[ \|\mathbf{h}(\mathbf{w})\|_2 \geq \inf_{u_1 > 0, u_1 < 0} \big\| \mathbf{h}(u_1 \bar{\mathbf{z}}_i + \mathbf{Q}_i' \mathbf{u}_i') - \mathbf{h}(u_2 \bar{\mathbf{z}}_i + \mathbf{Q}_i' \mathbf{u}_i') \big\|_2/2 \bigg| \mathcal{E}_i \bigg] \geq 1/2. \tag{D.3}$$

For any $u_1 > 0$ and $u_2 < 0$, denote $\mathbf{w}_1 = u_1 \bar{\mathbf{z}}_i + \mathbf{Q}_i' \mathbf{u}_i'$, $\mathbf{w}_2 = u_2 \bar{\mathbf{z}}_i + \mathbf{Q}_i' \mathbf{u}_i'$. We now proceed to give lower bound for $\|\mathbf{h}(\mathbf{w}_1) - \mathbf{h}(\mathbf{w}_2)\|_2$. By (D.2), we have

$$\|\mathbf{h}(\mathbf{w}_1) - \mathbf{h}(\mathbf{w}_2)\|_2 = \|a_i \mathbf{z}_i\|_2 \geq a_i/2, \tag{D.4}$$

where we use the fact that $\|\mathbf{z}_i\|_2 \geq 1/2$. Plugging this back into (D.3), we have

$$\mathbb{P}\big[ \|\mathbf{h}(\mathbf{w})\|_2 \geq |a_i|/4 \big| \mathcal{E}_i \big] \geq 1/2.$$

This completes the proof. $\qquad\square$