[Reviews · NeurIPS 2019]

Reviewer 1



While this paper makes a nice contribution to an important problem, I am not sure if it is significant enough for the conference. The overall outline of the analysis follows closely that of [2], and the main new component is the improved gradient lower bound, which is largely based on previous ones in [2] and [16]. Although the improved analysis provides new insight and I find it useful, I do not feel that it will provide a big impact. The other technical contribution on improved trajectory length is also nice but again I feel that it is somewhat incremental. The results seem technically sound; the proofs all look reasonable although I did not verify them thoroughly. The paper is well written and easy to follow in general, and the problem is an important one in machine learning. * After author response: I am satisfied with the response, but it does not change my opinion. Although the idea in the improved analysis is nice, I still feel that it is hard to push the idea much further, and very different ideas may be needed in order to improve the over-parameterization requirement significantly.

Reviewer 2



Originality: main technicalities including the assumptions, the initialization, and the main proof technique is very similar to the previous art. However, the improved upper bound in lemma 4.2. is insightful and novel, to the best of my knowledge. Quality: I only checked the proofs at a high level, but the submission seems sound to me, and the comparisons with the previous art seems legit. Clarity: The submission is very well-written and well-organized. The proof sketch section is specifically well done. Significance: the results are important because their improve the previous results on the over-parameterization requirements, however, the requirements are still far from being realistic. >>>>>>>>>>>>>>>>> after feedback <<<<<<<<<<<<<<<<<<<<<< I have read and considered the authors feedback in my final evaluation. I vote for accepting this paper.

Reviewer 3



The work describes the proof for establishing the milder over parameterization and better iteration complexity using GD or SGD under the assumptions of Gaussian random initialization for each layer, similar to 2 that results in great improvements to both. The assumptions and setup is similar to 2 (as referenced), with the main difference being the exploitation of the gradient regions for all the data points, that allow for larger area, tighter bounds and linear trajectory length. The complexity increase from this increased computation is stated to be O(n) for n data points. This needs to be factored into the complexity calculations and it is not apparent if this treatment was done justly under 4.1. Further, the remark under 3.11 sounds like it could be better explained for the case of 2 layer ReLU networks (than referring to appendix). It would also been great to include practical results for implementation using e.g. data sets to support the theory. Overall, this paper sounds like a positive contribution to advancing state of the art with above caveats.

[Author Response · NeurIPS 2019]

**To Reviewer #1**

**Q1: "I am not sure if this is significant enough...The improved analysis is useful but not provide a big impact"**

**A1:** Improving the over-parameterization condition from $\tilde{\Omega}(n^{24})$ to $\tilde{\Omega}(n^8)$ without making any additional assumption on the training data is by no means trivial. More importantly, our work clearly points out the right direction to improve the over-parameterization condition: (1) proving tighter gradient lower bound, and (2) proving shorter trajectory length for (stochastic) gradient descent. Our work sheds light on further improving the over-parameterization condition and trigger a lot of followup work to make the over-parameterization well-aligned with the neural network width used in practice. Moreover, we believe our proof idea can also be generalized to studying a broader class of neural networks (e.g., CNN, ResNet). Therefore, we believe our contribution to this line of research is significant and our work has a big impact in the field of deep learning theory.

**Q2: "...very different ideas may be needed, I do not see how the new insight can help improve the requirement..."**

**A2:** We don't fully agree with your comment on this point. Based on our current ideas, there is still a big room to improve the over-parameterization requirement. We explain it as follows: in our paper, the proved over-parameterization condition is $\tilde{\Omega}(n^8)$ when the gradient lower bound is roughly in the order of $O(m/n^2)$ (Here the other problem-dependent parameters are omitted, please refer to Lemma 4.1 for details). If we can improve the gradient lower bound to $O(m/n)$ or even $O(m)$ (in the same order as the gradient upper bound), the over-parameterization condition can be further improved to be $\tilde{\Omega}(n^5)$ or even $\tilde{\Omega}(n^2)$ respectively. This clearly shows the potential and promise of proving tighter gradient lower bound. To give you a more concrete idea, in the current paper, each "gradient region" is designed based on the minimum separation distance $\delta$ and has the same size. However, for high-dimension training data, the separation distance for some data points can be very large, which implies that some "gradient region" can actually has much larger size. Therefore, the total size of "gradient regions" can be greatly enlarged if the average-case separation distance rather than only the smallest one of all training data is taken into consideration. This can potentially lead to larger gradient lower bound. We will point it out in the future work section.

**To Reviewer #2**

**Q1: If the analysis for the case where the top layer weights are also updates.**

**A1:** Thanks for your positive and constructive comments on our paper. In our current submission, we did not optimize the top layer weights since we aim to make a fair comparison between our results and those in existing work (which don't optimize the top layer). However, we would like to emphasize that our analysis can be easily extended to the case when the top layer is optimized, and the resulting theoretical guarantees (over-parameterization condition and convergence rate) are at least the same as our results. The proof sketch is as follows: similar to our current proof, we can also define a small perturbation region around the initialization, but the new definition involves a constraint on the top layer weights. Then, it can be shown that the neural network enjoys good properties inside such region. Based on such good properties, we can prove that until convergence the neural network weights, including the top layer weights, would not escape from such region. Note that optimizing more parameter can lead to larger gradient, thus we can prove a larger gradient lower bound during the training process which can potential speed up the convergence of optimization algorithm (e.g., GD, SGD). Combining the above analyses, we can derive the corresponding over-parameterization condition and convergence rate, and note that these guarantees would be no worse than existing results presented in our paper. We will briefly discuss this extension in the final version.

**To Reviewer #3**

**Q1: "The complexity increase ... it is not apparent if this treatment was done justly under 4.1."**

**A1:** We apologize that we did not make it clear, but this is a misunderstanding on the exploitation of the gradient regions. We would like to clarify that exploiting gradient regions for all the data points is in the proof level, and *has nothing to do with the algorithm*. Therefore, it would never cause additional computation complexity for the algorithm (e.g., GD and SGD), and it will not affect the complexity calculations in our paper. In other words, the complexity calculations in our current paper are *correct*.

**Q2: " The remark under 3.11 sounds like it could be better explained for the case of 2 layer ReLU networks."**

**A2:** We are sorry for making you confused about Remark 3.11. In Remark 3.11, we would like to say that for 2-layer ReLU networks, we can derive its result from Theorem 3.8, because the result of Theorem 3.8 is for any depth $L \geq 2$, and therefore it is also applicable to 2-layer ReLU networks when choosing $L = 2$. However, we can prove stronger results specifically for 2-layer ReLU networks, by exploiting the special structure of 2-layer ReLU networks. Such stronger results are stated in Theorem 3.10. We use Remark 3.11 to compare the results in Theorem 3.8 when choosing $L = 2$, with the results in Theorem 3.11, and show what improvement we can achieve in Theorem 3.11. We hope this clarify your confusion.

**Q3: "It would also been great to include practical results for implementation ..."**

**A3:** Thank you for your suggestion. We will consider adding the experimental results in the appendix in the final version.

[Meta-Review · NeurIPS 2019]

This paper analyzes the convergence of GD and SGD for overparametrized networks, which result in improvements of the overparametrization requirement. Initially the paper received weakly positive reviews. Specifically, they felt that while the contribution follows closely prior work [2,16], the paper still makes a nice contribution, which is insightful and novel. The rebuttal addressed the issues raised by the reviewers. While one reviewer remained concerned as to whether the ideas in the paper can be extended further, upon discussion, the reviewers agreed that the paper should be accepted.